# Influence of Rain on the Abundance of Bioaerosols in Fine and Coarse Particles

Chathurika M. Rathnayake[1], Nervana Metwali[2], Thilina Jayarathne[1], Josh Kettler[1], Yuefan Huang[1], Peter S. Thorne[2,3], Patrick T. O'Shaughnessy[2,3], and Elizabeth A. Stone[1]

5   [1] Department of Chemistry, University of Iowa, Iowa City, 52242, United States.

[2] Occupational and Environmental Health, University of Iowa City, 52242, United States.

[3] Civil and Environmental Engineering, University of Iowa City, 52242, United States.

10   *Correspondence to*: Elizabeth. A. Stone (betsy-stone@uiowa.edu)

**Abstract.** Assessing the environmental, health and climate impacts of bioaerosols requires knowledge of their size and abundance. These two properties were assessed through daily measurements of chemical tracers for pollens (sucrose, fructose, and glucose), fungal spores (mannitol and glucans) and Gram-negative bacterial endotoxins in fine particulate matter ($PM_{2.5}$), coarse PM ($PM_{10-2.5}$) and $PM_{10}$ (as the combination of $PM_{2.5}$ and $PM_{10-2.5}$) during the spring tree pollen season (mid-April to early-May) and late summer ragweed season (late-August to early-September) in the Midwestern US in 2013. Under dry conditions, pollen and fungal spore tracers were primarily in coarse PM (>75%), as expected for particles greater than 2.5 µm. Rainfall on May 2 corresponded to maximum atmospheric pollen tracer levels and a redistribution of pollen tracers to the fine PM fraction (>80%). Both changes were attributed to the osmotic rupture of pollen grains that led to the suspension of fine-sized pollen fragments. Fungal spore tracers peaked in concentration following spring rain events and decreased in particle size, but to a lesser extent than pollens. A short, heavy thunderstorm in late summer corresponded to an increase in endotoxin and glucose levels, with a simultaneous shift to smaller particle sizes. Simultaneous increases in bioaerosol levels and decrease in their size has significant implications for population exposures to bioaerosols, particularly during rain events. Chemical mass balance (CMB) source apportionment modelling and regionally-specific pollen profiles were used to apportion PM mass to pollens and fungal spores. Springtime pollen contributions to $PM_{10}$ mass ranged from 0.04–0.8 µg m$^{-3}$ (0.2-38%, averaging 4%), with maxima occurring on rainy days. Fungal spore contributions to $PM_{10}$ mass ranged from 0.1–1.5 µg m$^{-3}$ (0.8–17%, averaging 5%), with maxima occurring after rain. Overall, this study defines changes to the fine and coarse mode distribution of PM, pollens, fungal spores, and endotoxins in response to rain in the Midwestern United States and advances the ability to apportion PM mass to pollens.

# 1 Introduction

Inhalable bioaerosols (<100µm) act as aeroallergens, triggering mild to severe allergic respiratory diseases (D'Amato et al., 2007a; Dales et al., 2003). Types of bioaerosols include viruses (<0.3 µm), bacteria (0.25-8 µm), fungal spores (1-30 µm), and plant pollens (~5-100 µm) (Jones and Harrison, 2004; Matthias-Maser and Jaenicke, 1995). Once inhaled, bioaerosols reach different regions of the respiratory system based on their size (Oberdörster et al., 2005; Brown et al., 2013), which is dependent on the route of breathing, age, gender, and activity level (Brown et al., 2013). In general, particles of 3 µm and 5 µm for adults and children, respectively, travel beyond the larynx (Brown et al., 2013). Human immune system produces antibodies against inhaled aeroallergens that initiate airway symptoms (e.g., cough and runny nose), and exacerbate diseases like asthma and allergic rhinitis. Allergic respiratory diseases are estimated to affect 334 million people worldwide, particularly children (GAN, 2014). These respiratory illnesses are predicted to increase in response to global trends of increasing carbon dioxide concentrations (Singer et al., 2005; Ziska and Caulfield, 2000) and temperatures (Beggs, 2004) that enhance the allergenicity (Singer et al., 2005) and quantity (Ziska and Caulfield, 2000) of pollens, and duration of pollen seasons (Beggs, 2004; Beggs and Bambrick, 2006). The protection of sensitive populations from bioaerosols requires understanding environmental exposures to bioaerosols as a function of their type, size, and temporal variation.

Ambient levels of pollens vary seasonally with plant phenology (Galán et al., 1995; Targonski et al., 1995). Springtime in the Midwestern United States is generally characterized by high levels of tree pollens (Targonski et al., 1995), such as oak (Wallner et al., 2009), birch (Emberlin et al., 2002), alder, and hazel (Niederberger et al., 1998). Summertime has elevated concentrations of grass pollens (e.g., Timothy and Rye grass) and weed pollens, especially ragweed (Targonski et al., 1995). Daily pollen levels are affected by temperature, with warmer conditions favouring pollen development, maturation, and active release (van Vliet et al., 2002). Rainfall promotes the passive release of intact pollens by agitation (Taylor and Jonsson, 2004). In rainy conditions, pollen grains absorb water, osmotically rupture, and release cytoplasmic starch granules (D'Amato et al., 2007b). Microscopy studies have shown that intact birch pollens of 22 µm in size can rupture and release around 400 starch granules (Staff et al., 1999) ranging from 0.03 - 4 µm (D'Amato et al., 2007b). Consequently, human exposures to pollens in the atmosphere are highly dependent on pollen type, season, and local meteorology.

Fungal growth and spore release is also promoted by elevated temperatures (Corden and Millington, 2001) and wet conditions (Pasanen et al., 2000). Fungi discharge spores via splash-induced emission, as is the case for *Cladosporium,* a prominent fungal genus (Troutt and Levetin, 2001; Oliveira et al., 2009) that releases spores by mechanical shock and fast air currents produced by rain drops (Elbert et al., 2007; Allitt, 2000). Fungi that belong to the division Ascomycetes disperse spores in moist conditions (Jones and Harrison, 2004) leading to elevated spore levels several hours after rain (Allitt, 2000; Packe and Ayres, 1985). The release of bioaerosols during and after rain events can trigger significant changes to ambient bioaerosol numbers (Knox, 1993; Huffman et al., 2013) and mass concentrations (Marks et al., 2001).

Bacteria in the atmosphere are typically attached to soil or vegetative surfaces as agglomerations of cells (Jones and Harrison, 2004). Taxonomic analysis has revealed that soil and plant surfaces serve as sources of bacteria in the Midwestern US (Bowers et al., 2011). Ambient bacterial levels increase with temperature (Carty et al., 2003) due to conditions that favour vegetation and bacterial habitat (DeLucca and Palmgren, 1986; Romantschuk, 1992). In vegetation covered areas, atmospheric bacterial concentrations have been shown to increase during and after simulated rain events (Graham et al., 1977; Robertson and Alexander, 1994) as well as natural rain events (Constantinidou et al., 1990; Huffman et al., 2013). This response to precipitation has been attributed to rain moving plants and aerosolizing bacteria (Jones and Harrison, 2004). With strong dependences on local meteorology, bacteria are likely to exhibit high temporal variability.

Once released, bioaerosols in the atmosphere promote cloud and ice nucleation (Pope, 2010; Sun and Ariya, 2006; Murray et al., 2012). Intact birch, walnut and willow pollens have been demonstrated to be cloud condensations nuclei (CCN) (Pope, 2010), with cytoplasmic pollens granules ranging 0.05-0.3 µm being the most CCN active, due to their hygroscopicity and longer residence time (Steiner et al., 2015). Bacteria also are CCN, at relatively low supersaturations (Sun and Ariya, 2006; Franc and Demott, 1998). Because of their ordered structures, bioaerosols are effective ice nuclei (IN) forming ice crystals at sub-cooled temperatures, including intact pollens (Diehl et al., 2001; Diehl et al., 2002), pollen extracts (Augustin et al., 2013), fungal spores and bacteria (Murray et al., 2012). Their ability to act as CCN and IN affects the earth's climate through changes to cloud albedo and precipitation cycles (Diehl et al., 2001; Sun and Ariya, 2006).

Atmospheric levels of bioaerosols can be assessed through measurements of specific chemical and biological tracers. Glucose, fructose and sucrose are main energy storage material in plants, major contributors to pollen mass (Speranza et al., 1997; Fu et al., 2012) and have been used as pollen tracers in China and the United States (Fu et al., 2012; Jia et al., 2010a; Jia et al., 2010b) Although not unique to pollens, these three sugars also comprise a minor fraction of suspended soil (Rogge et al., 2007), road dust (Simoneit et al., 2004) and biomass burning (Medeiros and Simoneit, 2008). Mannitol and arabitol are sugar alcohols that serve as energy storage materials in fungi and are used to identify the presence of airborne fungal spores and to quantify their contributions to PM mass (Bauer et al., 2008; Zhang et al., 2010). 1,3-β-D-glucans are immune-active polysaccharides in fungal cell walls (Thorn et al., 2001; Bonlokke et al., 2006) that are also tracers of fungal spores that have been used to assess exposure levels in indoor and outdoor environments (Madsen, 2006; Crawford et al., 2009). Endotoxins are lipopolysaccharides in Gram-negative bacterial membranes that induce respiratory inflammations (Douwes et al., 2003; Thorne et al., 2015). Ambient levels of endotoxins have been measured in outdoor (Pavilonis et al., 2013) and occupational settings (Thorne et al., 2009). Measurement of these bioaerosol tracers allows for the evaluation of the atmospheric concentrations and fine and coarse mode distributions of pollens, fungal spores, and Gram-negative bacteria. Given the important role of bioaerosols in the health of sensitive populations and in atmospheric processes, a robust understanding of bioaerosol types and their response to changing meteorological conditions is needed. Our central objectives were $i$) to assess temporal variations in pollens, fungal spores and endotoxin concentrations and their distribution across fine ($PM_{2.5}$) and coarse ($PM_{10-2.5}$) size modes, $ii$) evaluate environmental conditions including rain and temperature that lead to high bioaerosol levels and decreases in their size from $PM_{10-2.5}$ to $PM_{2.5}$, $iii$) determine intact pollen

diameters and chemically profile regionally-important pollen types (red oak, pin oak, cotton ragweed, giant ragweed and corn) for use in source apportionment, and *iv*) estimate pollen and fungal spore contributions to PM mass by way of chemical mass balance (CMB) modelling. The outcomes of this study include an improved understanding of changes in ambient bioaerosol concentrations and distributions across fine and coarse size modes in response to rain events and their

contributions to PM mass.

## 2 Methods

### 2.1 Sample collection

Daily (24 h) PM samples were collected from 17 April–9 May (springtime) and 15 August–04 September (late-

summer) in 2013, at the University of Iowa air monitoring site in Iowa City, Iowa, US (+41.6647, – 91.5845). The site was located at the University of Iowa Practice Fields in a suburban landscape in an open area surrounded by woods, agricultural fields, meadows and a parking lot. $PM_{2.5}$ and $PM_{10-2.5}$ were collected using an Andersen dichotomous sampler (Series 241) that included a $PM_{10}$ cut-off impactor (Anderson Instruments, Model 246b) and virtual impactor. The total air flow rate was 16.67 L min$^{-1}$ and the coarse flow rate was 1.667 L min$^{-1}$. PM samples were collected on 37-mm Teflon filters (Pall Corp.)

and $PM_{10}$ was determined as the sum of $PM_{2.5}$ and $PM_{10-2.5}$. The dichotomous sampler had a UMLBL (the University of Minnesota-Lawrence Berkeley Laboratory) type inlet which is equipped with a rain guard and a mesh-screen to exclude rain drops and insects. An additional set of $PM_{2.5}$ samples were collected on to 90-mm quartz fibre filters (Pall Life Sciences) using a medium-volume sampler (URG Corp.) equipped with a sharp-cut cyclone to select $PM_{2.5}$ at a flow rate of 90 L min$^{-1}$. Rain was excluded from the $PM_{2.5}$ sampler primarily by positioning the inlet downward and secondarily by the cyclone. Both

samplers were affixed to a platform 3 m above ground level and were unobstructed. Flowrates were measured using a rotameter at the beginning and the end of each sampling period; average flowrates were used to calculate air volumes. Filters were changed at 08:00 local time (CST) and one field blank was collected for every 5 samples. After sample collection, filters were stored at -20 ˚C in the dark.

To assess the representativeness of 2013 PM levels to typical conditions in Iowa, $PM_{2.5}$ and $PM_{10}$ mass

measurements were compared to measurements from 2010-2015 downloaded from the Technology Transfer Network (TTN) Air Quality System (AQS) Data Mart (USEPA, 2013). The federal reference method (FRM) site for Johnson County, Iowa is located at Hoover Elementary School, (+41.6572, – 91.5035), 6.3 km east of the University of Iowa air monitoring site. $PM_{2.5}$ concentrations were compared to average levels over the sampling period calculated from hourly measurements while $PM_{10}$ data were compared to filter measurements collected from midnight to midnight every three days.

### 2.2 PM mass measurement

PM mass was determined by the difference of pre- and post-sampling Teflon filter weights. Filter measurements made in a temperature (21.9 °C) and humidity controlled (25±5%) room using an analytical microbalance (Mettler Toledo

XP26) after conditioning 48 hours. Standard deviations of triplicate measurements were used as the error associated with the mass measurement.

### 2.3 Analysis of carbohydrates and inorganic ions

All glassware was prebaked at 500˚C for 5 hours, while plastic vials used were pre-rinsed with ultrapure (UP) water (resistivity >18.2 MΩ cm$^{-1}$) (Barnstead EasyPure II, 7401). Teflon filters (containing $PM_{10-2.5}$ samples) were cut in half using ceramic scissors on a clean, guided glass surface. Prior to extraction, Teflon filters were pre-wet with 100 µL of acetone (Sigma Aldrich). Subsamples of Teflon and quartz fibre filters (containing $PM_{2.5}$) were extracted into 4.00 mL of UP water by rotary shaking for 10 min at 125 rpm, ultra-sonication for 30 min at 60 Hz (Branson 5510, Danbury, CT, US), and then rotary shaking for 10 additional min. The extract was then filtered through a 0.45 µm polypropylene syringe filter (GE Healthcare, UK).

Carbohydrate concentrations were determined by high performance anion exchange chromatography (HPAEC) with pulsed amperometric detection (PAD, Dionex ICS 5000, Thermo Fisher, Sunnyvale, CA, USA). The HPAEC-PAD instrument consisted of an eluent organizer, dual pump, degasser, column compartment, electrochemical detector (ED50), AS-DV autosampler, CarboPac PA20 analytical column (3 X 150 mm, Dionex), guard column (3 X 30 mm), and a 10 µL injection loop. An isocratic separation of carbohydrates (erythritol, arabitol, fucose, trehalose [Alfa Aesar], glucose, fructose, arabinose, xylitol, xylose [Sigma Aldrich], rhamnose, mannose, ribose [Acros], sucrose and mannitol [Fisher Scientific]) was achieved with 10 mM sodium hydroxide (NaOH, Fisher Scientific) that was stored under $N_2$ (Praxair). The detector cell contained a gold disposable working electrode, to which quadruple waveform A was applied relative to a pH-Ag/AgCl reference electrode (Rocklin et al., 1998; Jensen and Johnson, 1997). Chromeleon 7 software was used for instrumental control, data acquisition and analysis. Carbohydrates were quantified against seven-point calibration curves ranging from 0.0100–2.50 ppm. Each analysis batch consisted of eight PM samples, two field blanks, one lab blank and one spike recovery sample. Summarized in Table S1 are carbohydrate extraction efficiencies (94-103%), instrument detection limits, and method detection limits.

Inorganic ion concentrations were determined using ion exchange chromatography with suppressed conductivity detection (ICS-5000, described above) following Jayarathne et al. (2014). Briefly, anions were separated on an Ionpac AS22 analytical column (4 X 250 mm, Dionex) preceded by a guard column and followed by a suppresser (Dionex AERS 500). Cations were separated on an Ionpac CS12A analytical column (3 X 150 mm, Dionex) preceded by a guard column, and followed by suppresser (Dionex CERS 500). Seven-point calibration curves were prepared from Seven Anion Standard and Six Cation-II Standard (Dionex) over the range of 0.010–10.0 ppm. Method performance metrics are summarized elsewhere (Jayarathne et al. 2014).

### 2.4 Analysis of biomarkers

Biomarkers were analyzed in extracts from the remaining halves of Teflon filters containing coarse PM and entire Teflon filters containing fine PM. Filters were extracted via shaking into 2 mL of sterile pyrogen-free (PF) water for 1 h at 22˚C. Extracts were then centrifuged (5 min at 600g at 4 ℃).

For analysis of fungal glucans, one aliquot of the supernatant was transferred into a PF borosilicate tube, mixed with 10x PF phosphate buffered Saline containing 0.05% Tween-20 (a surfactant), shaken for 1 h, autoclaved for 1 h, shaken for 20 min shaking, and then centrifuged for (600g at 4℃) 20min. Glucans were quantified by enzyme immunoassay as previously described by Blanc et al. (2005). A 12-point calibration curve prepared from (1-3, 1-6)-β-D-glucan (*scleroglucan*) ranged from 3-5000 ng mL $^{-1}$. The solution absorbance was measured at 450 nm (SpectraMax Plus 384; Molecular Devices, Sunnyvale, CA, USA).

For analysis of endotoxins, a second aliquot of the supernatant was subjected to the kinetic chromogenic *Limulus* amebocyte lysate assay (LAL) (Lonza, Inc., Walkersville, MD) as described in Thorne (2000). The 12-point calibration curve was generated utilizing endotoxin standard (*Escherichia coli* 055:B5) at concentrations ranging from 0.024-50 Endotoxin Units (EU) mL$^{-1}$. The solution absorbance was measured at 405 nm (SpectraMax M5, Molecular Devices).

## 2.5 Collection and analysis of pollens

Oak pollens were harvested from pin and red oak trees in park areas surrounding Iowa City during the spring of 2013 into pre-cleaned aluminium foil lined bags. Cotton and giant ragweed pollens were collected in late-summer of 2015 from bushes near roadways in residential areas of Iowa City. Cotton ragweed and corn pollens were purchased (Polysciences Inc., Warrington, PA). Pollen images were taken to determine pollen grain diameters using a Zeiss LSM 710 fluorescence microscope (Carl Zeiss Microscopy GmbH, 07745 Jena, Germany) following Pöhlker et al. (2012), and IX-81 inverted microscope (Olympus Corporation, Tokyo, Japan). Prior to extraction and chemical analysis, pollens were desiccated overnight and weighed (Mettler Toledo XS204 and XP26 balances). Pollens (~0.005–0.015 g) were extracted and analysed following the methods described in section 2.3.

## 2.6 Chemical Mass Balance (CMB) modeling

PM mass was apportioned to fungal spores and pollens using EPA-CMB model (version 8.2). $PM_{2.5}$ and $PM_{10}$ mass (from the sum of $PM_{2.5}$ and $PM_{10-2.5}$) was apportioned to bioaerosols using sucrose, glucose, fructose, and mannitol as fitting species. Input source profiles included one pollen profile selected from red oak, pin oak (this study), white birch, Chinese willow, or Peking willow (Fu et al., 2012) and one fungal spore profile (Bauer et al., 2008). Sensitivity tests were conducted to assess the fit of different pollen profiles to ambient measurements, focusing on sampling days from 26 April–9 May when pollen tracer levels were highest.

## 2.7 Statistical analysis

Prior to statistical analysis, data points below detection limits were substituted with the limit of detection (LOD)/ $\sqrt{2}$ (Hewett and Ganser, 2007). Concentration measurements were tested for normality and log-normality using the Anderson-Darling test in Minitab (version 16). Species concentration measurements were not normally distributed, thus Spearman's rank order correlation was employed for non-parametric comparisons ($r_s$) in Minitab (version 16). PM measurements were normally distributed thus t-tests comparing PM means from dry and rainy periods was conducted in Minitab (version 16). Significance was assessed at the 95% confidence interval ($p \leq 0.05$).

## 3 Results and discussion

Measurements of chemical tracers and biological markers are used to determine the relative concentrations and distribution of pollens, fungal spores, and bacteria in fine and coarse PM. Only few prior studies have combined chemical tracers and biological markers (Rathnayake et al., 2016; Chow et al., 2015), while many others have relied on either chemical tracers (Fu et al., 2012; Medeiros et al., 2006; Burshtein et al., 2011; Yttri et al., 2007; Zhang et al., 2010) or biological assays (Nilsson et al., 2011; Mueller-Anneling et al., 2004; Pavilonis et al., 2013; Madsen et al., 2011; Singh et al., 2011). Glucose, fructose, and sucrose are major components of pollens, mannitol and fungal glucans are in fungal spores, and endotoxins are in bacteria. In the ambient particulate matter, these species are used as bioaerosol tracers, since their concentrations reflect mass concentrations of the corresponding bioaerosol. These species provide general insight to classes of bioaerosols present, but cannot be used for species-level identification, which requires either microscopy imaging or DNA sequencing.

### 3.1 Characterization of pollens common to the Midwestern US

Red oak, pin oak, corn, cotton ragweed and giant ragweed pollen ranged in average diameter from 20 - 90 µm (Figure S1, Table 1). Together, glucose, fructose and sucrose accounted for an average of 5–14 % of pollen mass, while erythritol, arabinose, mannitol and rhamnose were detected in trace amounts (Table 1). Due to the relatively high mass fraction of glucose, fructose, and sucrose in pollens in the present and in prior studies (Fu et al., 2012; Speranza et al., 1997) these carbohydrates are the best candidates for assessing pollen contributions to ambient PM. Notably, the carbohydrate distributions in corn pollens differ from those previously reported (Speranza et al., 1997), with differences likely resulting from genetics (Speranza et al., 1997) and environmental factors (e.g. temperature, availability of water, and $CO_2$ levels) that are known to affect the synthesis and storage of carbohydrates (Aloni et al., 2001; Yoshida et al., 1998; Vesprini et al., 2002). Across different pollen types, the relative abundances of glucose, fructose and sucrose varied. For instance, the most abundant carbohydrate was sucrose for red oak, pin oak, and Polysciences cotton ragweed, fructose for corn pollen, and glucose for local cotton and giant ragweed. Sucrose to fructose ratios across different pollen types may serve to identify pollen types in ambient PM, in cases when a single pollen type is dominant (as discussed in section 3.6.1).

### 3.2 Fine and coarse PM concentrations

### 3.2.1 Spring

From 17 April to 9 May, 2013, daily $PM_{10}$ levels in Iowa City ranged from 2–32 µg m$^{-3}$ (with an average of 15±8.9 µg m$^{-3}$), and fine PM ranged from 2–13 µg m$^{-3}$ (with an average of 7.1±3.0 µg m$^{-3}$). Comparison to PM levels at a nearby FRM site (located 6.3 km to the east) from 2010-2015 (Table S2), demonstrated that spring 2013 PM levels were typical for the surrounding years.

On 15 of the 23 spring sampling days, conditions were dry and no rain occurred (Figure 1a). On the remaining 8 days, daily rainfall totalled 0.3–85 mm. Rainfall corresponded to low PM concentrations with average fine PM levels decreasing from 8.3±2.6 µg m$^{-3}$ on dry days to 4.7±2.2 µg m$^{-3}$ on rainy days and coarse PM levels decreasing from 10±5.6 µg m$^{-3}$ to 1.9±1.5 µg m$^{-3}$ (Figure 1b). The PM reduction on rainy days was statistically significant ($p<0.01$) and was driven by wet deposition of PM in both size modes. Rain also affected the distribution of particles between the fine and coarse modes. $PM_{2.5}$ contributed 48±11 % of $PM_{10}$ on rainy days compared to 80±13 % on dry days. The shift in the PM size from coarse to fine modes reflects that rain was more effective at scavenging and/or suppressing the release of coarse particles compared to fine particles. This is consistent with previous ambient studies that have demonstrated coarse PM is more effectively scavenged than fine particles (Guo et al., 2016; Li et al., 2016). Particle removal via rainfall depends on many factors including a strong dependence on the particle size (Gregory, 1962; Baklanov and Sørensen, 2001); airborne particles with diameters greater than 3 µm have a higher tendency to collide with falling rain drops and are effectively scavenged via inertial impaction (Wang et al., 2010; Andronache, 2003; Mircea et al., 2000).

**3.2.2 Late summer**

Only one brief rain occurred during the three-week campaign, on August 22 when a thunderstorm brought 1.0 mm between 10–11 am (Figure 2a). From 15 August to 4 September, 2013, Iowa City daily $PM_{10}$ levels as shown in Figure 2b, ranged from 21–50 µg m$^{-3}$ (averaging 33±8 µg m$^{-3}$) and fine PM levels ranged from 3–17 µg m$^{-3}$ (averaging 12±4 µg m$^{-3}$). On average, fine PM accounted for 39±12 % of $PM_{10}$. Compared to adjacent years (2010-2015), the late-summer of 2013 exhibited higher PM levels (Table S3). This is attributed to unusually dry conditions that reduce soil moisture leading to increase soil resuspension, and lack of wet deposition.

**3.3 Pollen tracers**

**3.3.1 Spring**

The temporal variations of pollens were assessed utilizing the combination of glucose, fructose and sucrose as chemical tracers. Ambient concentrations of these pollen tracers were relatively low from 17–25 April when lower temperatures (averaging 7 ˚C) and rainy conditions prevailed. Pollen tracer levels were relatively higher from 26 April–9 May, coinciding with warmer temperatures (averaging 15 ˚C) that marked the transition from winter to spring (Figure 1c-e, Table S4). Temperature and coarse mode glucose and sucrose were significantly correlated ($r_s \geq 0.8$, $p<0.001$), reflecting that warmer temperatures promote the development, maturation, and release of pollens.

After the onset of spring, rain events increased pollen levels. For instance, maximum fructose and sucrose levels occurred on 2 May and maximum glucose on 9 May; rain occurred on both of these days, following a dry period with relatively high temperatures. Remarkably, rain events substantially altered the fraction of pollen tracers in fine and coarse modes. On a typical dry day, more than 80% of pollen tracers were present in coarse PM, which is expected for pollen particles that have geometric diameters in the range of 5-100 µm (Huffman et al., 2010). However, when pollen markers peaked on 2 May, mass fractions of glucose, fructose and sucrose in the fine mode reached 83%, 91% and 93%, respectively (Figure 1c–e, right axis). With continued rainfall on 3–4 May, pollen markers remained elevated in the fine mode relative to coarse PM. After the rain stopped, coarse mode pollens increased in concentration and resumed the typical distribution across fine and coarse modes by 5 May. Light rainfall on 9 May coincided with increases in glucose in both size modes, with only 14% of these tracers in the fine mode. Together, these data suggest release of pollen fragments less than 2.5 µm during some rain events (2–4 May) and the passive release of some pollen particles in the coarse particle size range during others (9 May). Notably, this is the first observation of the release of fine particle pollen fragments to the atmosphere using chemical tracers. Most field measurements include analysis of either $PM_{2.5}$ or $PM_{10}$, while measurements in both size modes are required to capture this phenomenon.

The likely explanation for the increase in airborne pollens and simultaneous decrease in their size on May 2 is the rupturing of pollen walls as a result of the osmotic pressure that builds up inside the pollen due to absorbed moisture during rain (Taylor et al., 2004; Taylor et al., 2002). Osmotic shock has been previously demonstrated to cause rupturing of grass and birch pollens that releases cytoplasm (Taylor et al., 2004; Taylor et al., 2002; Suphioglu et al., 1992). Gusty winds can loft pollen fragments (Wallis et al., 1996) and strong winds on 2 May are likely to have contributed to the elevated fine pollen levels.

Differences in the distributions of pollen tracers across fine and coarse modes during the rain events on 2 May (mostly fine PM) and 9 May (mostly coarse PM) are expected to result from different pollen types predominating as evidenced by differing ratios of carbohydrates. On 2 May, the relative ratios of glucose and sucrose (normalized to fructose) in fine PM were 1.4 and 2.5, respectively, close to the ratios of red oak (1.2 and 2.1, respectively). Oak trees are abundant in Eastern Iowa and a prominent pollen type in the springtime, making oak a likely (but unconfirmed) source of pollens in fine PM. Meanwhile, the respective carbohydrate ratios on 9 May (18 and 0.7, respectively) did not match any of the local or literature available pollen profiles. These data suggest that certain pollen types undergo osmotic rupturing and release fine particles, while others do not. Further studies are needed to identify the types of pollens that rupture and conditions under which osmotic rupturing occurs.

### 3.3.2 Late summer

From mid-August to early-September, average temperature was moderately correlated with coarse mode glucose, fructose and sucrose ($r_s > 0.5$, $p < 0.02$). In the fine mode, glucose was frequently detected, while fructose and sucrose not (Figure 2c-e); this is likely due to the predominant pollen type having higher glucose concentrations relative to fructose and

sucrose, as is the case for ragweed pollens (Table 1). The potential of glucose deriving from soil (Rogge et al., 2007; Simoneit et al., 2004) suspended in the air by splashing (Joung and Buie, 2015) was eliminated because there was no corresponding change in calcium, a well-established soil-tracer. Consequently, glucose is considered to be a tracer for pollens even in the absence of the other two pollen tracers, and the discussion of pollen distribution across fine and coarse

PM relies solely on glucose for this time period. On average, 83% of glucose mass concentration was found in coarse mode (Figure 2c), consistent with typical size range of intact pollens (Huffman et al., 2010). The single late-summer rain event on 22 August coincided with an increase in fine mode glucose concentration and an increase of the fine PM fraction of glucose to 34%, compared to 16% on dry days. The late summer single rain event indicated passive release of pollen fragments in response to rain that was similar to spring (section 3.3.1). However, with only one rain event occurring in the late summer

study in 2013, additional studies are needed to validate these trends and identify the responsible pollen types.

### 3.4 Fungal spore tracers

### 3.4.1 Spring

Daily coarse mode fungal spore tracer concentrations significantly correlated with daily average temperature: fungal

sugar mannitol and temperature ($r_s$=0.7, p<0.001) and the fungal cell wall component glucan and temperature— ($r_s$=0.4, p=0.04). From 17–21 and 23-25 April, cooler temperatures prevailed (averaging 6 and 7 ˚C, respectively) and $PM_{10}$ mannitol and glucan concentrations were relatively low (Figure 3a and b). An exceptionally high $PM_{10}$ glucan level occurred (Figure 3b) on April 22, when temperature increased to a local maximum of 14 ˚C. From 26 April, temperatures warmed to an average of 15 ˚C, concurrent with an increase fungal spore tracer levels. The correlation of temperature with fungal spore

tracers is consistent with warmer temperatures favouring fungal growth (Corden and Millington, 2001; Rodriguez Rajo et al., 2005). The two tracers were moderately correlated with one another ($r_s$=0.5, p<0.02), signifying their origin from the same source.

Rain influenced ambient concentrations and the fine and coarse mode distributions of fungal spore tracers, likely by triggering passive and/or active release mechanisms and/or promoting fungal growth. Maximum mannitol and glucan levels

occurred on 5 May, which followed three days with rain (Figure 3a-b). Rainfall facilitates fungal growth promoting fungal germination and hyphal growth (Schulthess and Faeth, 1998; Morris et al., 2016) and wet conditions that follow rain are favourable for active release of fungal spores (Rodriguez Rajo et al., 2005; Van Osdol et al., 2004). For instance, actively discharged ascospores peak after rain in wet conditions (Troutt and Levetin, 2001; Elbert et al., 2007; MacHardy and Gadoury, 1986). Fungal spore tracer levels in coarse PM dropped on days when rain fell (e.g. 23 April, 2 May), due to

particle removal by wet deposition. The fine and coarse mode distributions of fungal spores, which typically have intact diameters in the range of 1–30 µm (Jones and Harrison, 2004), also were influenced by rain.. During dry days, 13% of fungal spore tracers were in the fine PM fraction. On rainy days, the fraction of fungal spore tracers in the fine mode reached local maxima at 41% (23 April), 36% (24 April), and 54% (2 May) for mannitol and 38% for glucans (23 April; Figure 3a-b, right axis). The relative decrease in the size of fungal spores is attributed to a combination of the passive release of fungal spores

less than 2.5 µm via rain splash and mechanical agitation of vegetative surfaces by rain drops (Allitt, 2000; Elbert et al., 2007; Huffman et al., 2013), and the removal of coarse fungal spore particles by droplet scavenging. Compared to pollens (section 3.3.1), rain events impacted the fine and coarse mode distributions of fungal spores to a much lesser extent.

**3.4.2 Late summer**

From mid-August to early-September atmospheric concentrations of mannitol correlated with temperature ($r_s$= 0.5, p=0.01), consistent with increased fungal growth with elevated temperatures (section 3.4.1). Fine mode mannitol reached a maximum on 22 August when rain fell during a one hour period (Figure 4a), likely due to fungal spore release via rain splash and mechanical agitation (section 3.4.1). Coarse mode mannitol also increased on 22 August, most likely due to release of fungal spores after rain subsided in response to wet conditions. Mannitol in fine PM accounted for an average of 9±4 % of the total $PM_{10}$ concentration and was not substantially different on 22 August (14%).

Coarse mode glucan concentrations in late summer were neither correlated with temperature ($r_s$=0.01, p=1), nor mannitol ($r_s$=0.2, p=0.3). Mannitol concentrations and fungal spore counts have spatial and seasonal differences from one another (Bauer et al., 2008), likely due to differences in mannitol emission per spore across fungal types (Elbert et al., 2007; Bauer et al., 2008) and/or mannitol concentrations in spores from within a species (e.g. ascomycetes releases ascospores during sexual reproduction and conidia during asexual reproduction (Nauta and Hoekstra, 1992)). The glucan content in fungal cell walls also vary with the fungal species (Foto et al., 2004). Collectively, these differences could give rise to weak or negligible correlations of ambient mannitol and glucan concentrations. Alternatively, non-fungal sources of either mannitol or glucans would confound their correlation. For instance, higher plants and some algae contain mannitol in their structure (Loescher et al., 1992; Shen et al., 1997). Ragweed pollens contain glucans (Foto et al., 2004), is a possible glucan source in late summer when ragweed pollens are prevalent and glucans significantly correlate with sucrose ($r_s$=0.5, p=0.04). Alternatively glucans may have derived from bacterial cells (McIntosh et al., 2005; Rylander and Lin, 2000), even though their correlation was not significant ($r_s$=0.4, p=0.1). Although glucans appear to have been influenced by bacterial and pollen levels in addition to fungi, the assessment of their ambient concentrations remains important, because they are immunostimulants that negatively impact human health (Thorn, 2001; Bonlokke et al., 2006).

**3.5 Bacterial endotoxins**

**3.5.1 Spring**

Coarse mode bacterial endotoxins, measured in endotoxin units (EU) against an *Escherichia coli* (055:B5) standard, were significantly correlated with daily average temperature ($r_s$=0.7, p<0.001). Lower temperatures averaging 7 ˚C from 17-25 April, led to low endotoxin levels compared to a warmer period averaging 11-23 ˚C from 26 April-1 May. The correlation of endotoxins with temperature agrees with prior ambient studies (Carty et al., 2003; Guan et al., 2014; Degobbi et al., 2011; Rathnayake et al., 2016) and is attributed to warmer temperatures increasing vegetative surfaces that serve as substrates for bacterial growth (Romantschuk, 1992; DeLucca and Palmgren, 1986; Carty et al., 2003). Heavy rain on 2 and 3 May led to a

drop in $PM_{10}$ endotoxin concentrations, due to wet deposition and suppression of soil dust particles upon which bacteria settle. On average, $92\pm5$ % of $PM_{10}$ endotoxins were in the coarse mode (Figure 3c). The distribution of bacterial endotoxins as well as bacterial cells towards larger particles has been demonstrated previously (Nilsson et al., 2011; Monn et al., 1995; Shaffer and Lighthart, 1997). Such observations reflect the association of bacteria with particles prominent in coarse mode

5   such as plant parts, animal parts, soil, spores or pollen surfaces (Jones and Harrison, 2004; Shaffer and Lighthart, 1997). In addition, it has been suggested that bacteria settled on particles are more likely to survive in the atmosphere compared to a single bacterium (Lighthart et al., 1993). Coarse mode endotoxins demonstrated a moderate positive correlation with calcium, the crustal element ($r_s$=0.7, $p<0.001$), which suggests soil resuspension as a source of endotoxins in Iowa City, which has been demonstrated previously in the Midwestern US (Bowers et al., 2011; Rathnayake et al., 2016).

### 3.5.2 Late summer

In late summer, ambient endotoxin concentrations had a positive moderate correlation with coarse mode endotoxins ($r_s$=0.5, $p$=0.02) similar to springtime (section 3.5.1). On 22 August, the only late summer day with rain, fine mode endotoxin concentrations reached a maximum (Figure 4c). Meanwhile, the endotoxin fraction in the fine mode

increased to 36% relative to an average of 5% on dry days. Rainfall promotes bacterial growth, such as *Pseudomanas syringae* that are common on plant surfaces and rapidly increase their populations during raining (Hirano and Upper, 1990; Hirano et al., 1996). The release of endotoxin to fine PM is expected to be caused by the aerosolization of Gram-negative bacteria living on plant surfaces (e.g., *Pseudomanas syringae, Pseudomanas fluorescens,* and *Pseudomanas viridiflava etc*. (Murray et al., 2012)) by agitation of plants or fungi by falling rain (Jones and Harrison, 2004; Constantinidou et al., 1990).

Soil resuspension was suggested as an important source of bacterial endotoxins in spring (section 3.5.1), however coarse mode endotoxins were not significantly correlated with calcium in late summer ($r_s$=0.2, $p$=0.33), suggesting that this is not the case. Consequently, non-soil bacterial sources were likely responsible, such as plant surfaces (Romantschuk, 1992; Jeter and Matthysse, 2005; Murray et al., 2012) that are probably agricultural row crops (Lindemann et al., 1982; Hirano et al., 1996) in the agricultural state of Iowa. This link could be further explored by examining the co-occurrence of bacterial

endotoxins with markers of plant waxes (i.e. odd-numbered *n*-alkanes), but is beyond the scope of the present study. The comparison of spring and late-summer endotoxin behaviour in response to rain suggests that soil bacteria are dominate in springtime, while bacteria residing on plant surfaces dominate in late-summer.

### 3.6 Contributions of pollens and fungal spores to PM mass

CMB source apportionment modelling was applied to estimate mass contributions of pollens and fungal spores to $PM_{10}$ and $PM_{2.5}$. This work extends the application of fungal spores tracer-to-mass ratios to estimate their contributions to PM mass (Di Filippo et al., 2013; Zhang et al., 2010) to pollens for the first time. The CMB model requires representative source profiles for sources, which were drawn from the literature in the case of fungal spores (Bauer et al., 2008), birch, and willow pollens (Fu et al., 2012), and from this study (section 3.1).

### 3.6.1 Source apportionment in spring

The pollen profiles that explained the greatest fraction of the variance in the springtime measurements (assessed by the CMB $R^2$ value) were pin oak and red oak (Figure S2). The resultant $R^2$ value further increased when fungal spores were added to the model (Figure S2). Birch and willow profiles, which showed an excess of sucrose (Fu et al., 2012) explained a substantially lower fraction of the variance in ambient data, where glucose and fructose concentrations outweighed sucrose. Hence, birch and willow pollen profiles were not considered further. Model results from using pin oak or red oak profiles in concert with the fungal spore profile produced consistent source contributions that were strongly correlated (Figure S3). Because red oak and pin oak fit ambient data to a comparable extent and both are sources of atmospheric pollens in Iowa, the best estimate of pollen contributions was calculated as the average contribution from red oak and pin oak.

Pollen and fungal spore contributions to $PM_{10}$ and $PM_{2.5}$ estimated by the CMB model are shown in Figure 5 (and Table S6). Overall, contributions to fine PM after onset of spring, from 26 April-09 May ranged from 0.01–0.7 µg m$^{-3}$ for pollens and 0.03 – 0.1 µg m$^{-3}$ for fungal spores, while contributions to $PM_{10}$ were consistently higher at 0.04–0.8 µg m$^{-3}$ for pollens and 0.13–1.5 µg m$^{-3}$ for fungal spores. On dry days, pollens contributed an average of 0.7% of $PM_{2.5}$ and 3.3% of $PM_{10}$. On rainy days, pollen contributions to fine PM averaged 11% and reached a maximum of 42% on May 2. Fungal spore contributions to fine PM averaged 0.5% on dry days and 1.7% on days with rain. Meanwhile, fungal spores had the greater contributions to $PM_{10}$ mass on days following rain, reaching 8.7% on May 5. These source apportionment results demonstrate that bioaerosol contributions to $PM_{10}$ mass in spring are typically low with averages of 4% and 5% for pollens and fungal spores, respectively), but can be significantly greater on days with rain, when bioaerosols are released and PM is removed by wet deposition. The distribution of bioaerosols in fine and coarse PM during spring is shown in Figure 6. For dry conditions, ~11% of pollens and fungal spores were observed in fine PM. However, during rainy days, 62% of pollen mass and 20% of fungal spore mass were observed in fine PM. These results indicate the importance of rain altering fine and coarse mode distribution of bioaerosols by affecting release mechanisms (i.e. passive release by splashing and mechanical agitation, or osmotic rupture of pollens).

Bioaerosol contributions to PM in this study were relatively in good agreement with prior studies. The average fungal spore contribution to $PM_{10}$ in spring (5%) was 1.6 times higher than suburban site of Vienna, Austria, and 1.6 times lower than a tropical rainforest in China (Zhang et al., 2010), which were measured during springtime. Collectively, contributions from pollens (3.3%) and fungal spores (0.9%) to fine PM was ~2 times lower than contributions reported in US which determined in summertime (Coz et al., 2010). The slight variations of contributions could be attributed to the differences in ambient bioaerosol levels and geographical differences.

### 3.6.2 Source apportionment in late-summer

PM mass could not be apportioned to pollens in late summer, because of poor agreement between ambient data and source profiles. Fewer than 10% of the ambient PM samples had relative ratios of sucrose, fructose, and glucose in the range

ragweed pollen profiles, which is a dominant pollen type in the Midwest. This lack of agreement could result from mixtures of pollen in the atmosphere that are not represented when utilizing a chemical profile for a single pollen type, and/or other dominant pollen types during late summer (e.g. Timothy grass and rye grass).

Fungal spore contributions to PM were estimated using the average mannitol conversion factor of 1.7 pg mannitol spore$^{-1}$ (range from 1.2-2.4 pg mannitol spore$^{-1}$) and a spore mass of 33 pg from Bauer et al. (2008). Resultant fungal spore mass contributions to $PM_{2.5}$ and $PM_{10-2.5}$ ranged from 0.04-0.31 µg m$^{-3}$ and 0.45-3.44 µg m$^{-3}$, respectively, (Table S7). The contribution of fungal spores to $PM_{2.5}$ averaged 1% on dry days, and 3% on 22 August when it rained. Meanwhile, fungal spore contributions to $PM_{10-2.5}$ averaged 6% and reached to 16% on 22 August. The maximum fungal spore contributions to PM on 22 August is likely due to fungal spores released during rain by passive mechanisms and after rain by active mechanisms (section 3.4.1). This leads to an increase in fine sized fungal spores when raining, and coarse sized spores post-rain (Huffman et al., 2013; Hjelmroos, 1993).

### 3.7 Implications of the release of fine bioaerosols surrounding rain events

The release of fine sized bioaerosols can influence cloud formation, by acting as CCN and IN. Pollen fragments are effective CCN and IN (Pope, 2010; Diehl et al., 2001). During rain intact pollen particles can swell and rupture, producing hundreds of fine-sized pollen particles (D'Amato et al., 2007b), significantly increasing the number of CCN and IN active particles in the atmosphere. Bacteria and fungal spores also active IN and CCN (Murray et al., 2012; Sun and Ariya, 2006; Hassett et al., 2015). Bacterial strains with higher IN activity (mostly Gram-negative bacteria that habitat plant surfaces (Murray et al., 2012), such as *Pseudomanas syringae*) increase in population during rain (Hirano et al., 1996), which can substantially increase airborne IN (Morris et al., 2016) that can persist in the atmosphere for weeks following rain (Bigg et al., 2015). Rainfall in general favours fungal growth (Schulthess and Faeth, 1998; Morris et al., 2016) as well as passive and active release of spores (Rodriguez Rajo et al., 2005; Van Osdol et al., 2004; Allitt, 2000; Elbert et al., 2007; Huffman et al., 2013) thereby increasing CCN and IN active particles in the atmosphere. When decreased in size (< 2.5 µm), these bioaerosols are more effective IN (Murray et al., 2015; Huffman et al., 2013). Because smaller particles have longer atmospheric lifetimes, fine bioaerosols will be transported longer distances before deposition, and thus may have effects in areas downwind of their release.

In general, the release of pollens, fungal spores, and Gram-negative bacteria in fine particles during rain events in Iowa, have the potential to influence human health. Elevating ambient fungal spore levels, particularly from species like *Penicillium, Aspergillus* and *Cladosporium,* trigger allergenic respiratory diseases like allergic rhinitis and asthma (Garrett et al., 1998; Tillie-Leblond et al., 2011; Knutsen et al., 2012) and high environmental exposures may lead to asthma exacerbations (Dales et al., 2003). Likewise, endotoxins induce inflammations in the respiratory tract (Dales et al., 2006; Liebers et al., 2008; Thorne et al., 2015). When pollen levels increase in concentration and decrease in size (as observed on May 2, May 9, and August 22), likely due to pollen rupturing, cytoplasmic pollen allergens (Suphioglu et al., 1992; Grote et al., 2001) will be released, leading to more direct exposure of humans to aeroallergens through inhalation. In the form of

smaller particles, aeroallergens penetrate deeper into the respiratory tract where they may trigger more severe allergenic responses (Taylor et al., 2002; Wilson et al., 1973). Acute asthma epidemics have been associated with rain events have been documented in Australia, Europe, Mexico and the US (D'Amato et al., 2016; Dales et al., 2003; Grundstein et al., 2008) earning the name "thunderstorm asthma." Such epidemics typically occur during pollen seasons (D'Amato et al., 2007a;

D'Amato et al., 2007b; D'Amato et al., 2016) and have been associated with ambient pollen counts (Marks et al., 2001). While lightning is associated with tropospheric ozone formation (Griffing, 1977; GAN, 2014) lightning alone (in the absence of rain) has not caused asthma epidemics (Grundstein et al., 2008), suggesting that rainfall plays an important role in thunderstorm asthma.

Pollen forecasting models currently do not include mechanisms for the release of pollen in response to rain and

instead assume that rain serves only as a sink of pollens, by means of droplet scavenging and wet deposition (Zhang et al., 2013). This erroneous assumption leads to predictions of low atmospheric pollen levels on days with rain (e.g. May 2), when pollen tracer levels are highest and primarily in the form of fine particles. A more accurate representation of airborne pollen levels is needed to support an early-warning system to sensitive populations, but must go beyond simply the co-occurrence of elevated pollen levels and thunderstorms, which are suggested to cause too many false alarms (Newson et al., 1998). For

accurate model parameterizations, a mechanistic and species-level understanding of pollen bursting is needed and should include definitions of the pollen types, seasonality, and meteorological conditions that promote the release of fine pollen particles to the atmosphere. In the meantime, persons suffering from pollen allergies should follow the recommendations of D'Amato et al. (2007b): "when asthmatic patients realize that a thunderstorm is approaching, the best thing for thing for them to do is to stay indoors, with windows closed."

The results of this study provide new insight and tools to better understand the potential scope of thunderstorm asthma. While thunderstorm asthma has been documented in several locations, the data presented herein provide the first evidence of this phenomenon occurring in the Midwestern US. Thunderstorms and heavy rain are common in this region during spring, and thus it is anticipated that conditions characteristic of thunderstorm asthma likely occur several times annually. Pollen prediction indices do not currently account for

the release of fine pollen fragments during rain, and consequently sensitive populations are not forewarned. To understand the potential for conditions that trigger thunderstorm asthma more broadly, chemical tracer approaches, as used here, are a useful tool. Chemical tracers provide a sensitive method of detecting fine pollens particles that may be useful in monitoring conditions that precede $PM_{2.5}$ pollen release. Because carbohydrates are not expected to undergo chemical alternation by the pollen bursting, they also provide a means of tracking

pollens across PM size fractions and associating pollens with their species of origin. Microscopy-based methods are challenged by changes to particle size and morphology upon bursting, which may require use of multiple microscopy techniques suitable for different particle sizes. Chemical tracer methods have potential to be broadly

applied, as national monitoring programs routinely collect PM$_{2.5}$ samples on filters for chemical analysis. In this way, regions and atmospheric conditions that lead to high levels of PM$_{2.5}$ pollen particles may be better defined.

## 4. Conclusions

Daily concentrations of PM mass and bioaerosol tracers (including fructose, glucose, and sucrose for pollens, mannitol and glucans for fungal spores, and endotoxins from Gram-negative bacteria) demonstrated high day-to-day variability and influences from meteorology, particularly rain. Elevated bioaerosol tracer levels were observed when temperatures are warmer suggesting increased pollen, fungal and bacterial concentrations during both spring and late summer periods. Rain events of spring triggered the release of pollens, with maximum levels of pollen tracers occurring on

May 2 and May 9, when rain occurred following a period of elevated temperatures in spring. Airborne fungal spore tracers in coarse PM fraction, however, were suppressed by spring rain and increased in concentration following rain events. Source apportionment by CMB modelling in concert with Midwestern pollen profiles indicated significant contributions from bioaerosols to PM mass on rainy days during springtime. Importantly, the fine and coarse mode distributions of endotoxins, pollen and fungal spore tracers shifted towards fine particles (<2.5 µm) during periods of rain. The fragmentation of pollens

due to osmotic rupture, shown previously through microscopy methods. For the first time, we demonstrate a shift of coarse particle pollens (2.5-10 µm) to fine particles (2.5 µm ) by way of chemical tracers during a major rain event and propose that this is due to osmotic rupture of pollens. The release of finer-sized bioaerosols during rain events has important implications for human exposures, because finer particles may penetrate more deeply into the lung and be transported over longer distances.

A detailed level of understanding of pollen release mechanisms, particularly as pollen fragments, is needed to improve the accuracy of allergen prediction models that erroneously forecast low airborne allergen levels during periods of rain. Future research should focus on a more precise determination of the duration of heightened pollen levels during rain events with higher time resolution measurements. Similarly, measurements with higher PM size resolution should be employed to determine the specific size range of pollen fragments during these events. Additional efforts are needed to

characterize the fungal and floral species that release fine-sized bioaerosols to the atmosphere and the mechanisms that trigger such release, to allow for their accurate representation in atmospheric models to support accurate representations of environmental conditions and forewarn susceptible populations of conditions that may lead to high bioaerosol exposures.

**Acknowledgements**

We thank Ralph Altmaier and Lindy Carr for their assistance with PM sample collection and gravimetric analysis and Prof. Keri Hornbuckle for establishing the University of Iowa Air Monitoring Site. We also thank Jianqiang Shao and Katherine Walters for helping with fluorescence and inverted microscope images, and the University of Iowa Central Microscopy Research Facility, a core resource supported by the Vice President for Research & Economic Development, the Holden Comprehensive Cancer Center and the Carver College of Medicine. This research was supported by the

Environmental Health Sciences Research Center (EHSRC) seed grant program (NIH P30 ES005605) and the University of Iowa.

Table 1: Pollen diameter and mass fractions of carbohydrates and ions with standard errors. The carbohydrates arabitol, xylitol, trehalose, fucose, mannose, xylose and ribose were below detection limits.

| | Red Oak | | | Pin Oak | | | Corn | | | Cotton ragweed[a] | | | Cotton ragweed[b] | | | Giant ragweed[b] | | |
|---|---|---|---|---|---|---|---|---|---|---|---|---|---|---|---|---|---|---|
| **n** | 5 | | | 5 | | | 5 | | | 5 | | | 3 | | | 3 | | |
| **Diameter (µm)[c]** | 30 | | | 30 | | | 80 | | | 20 | | | 35 | | | 35 | | |
| **Carbohydrates (µg mg$^{-1}$)** | | | | | | | | | | | | | | | | | | |
| Glucose | 41.1 | ± | 4.1 | 40.2 | ± | 3.5 | 15.2 | ± | 0.9 | 15.9 | ± | 1.6 | 43.3 | ± | 2.0 | 39.2 | ± | 2.8 |
| Fructose | 33.0 | ± | 1.8 | 33.9 | ± | 2.9 | 25.0 | ± | 1.1 | 13.5 | ± | 0.6 | 24.4 | ± | 1.1 | 22.9 | ± | 1.5 |
| Sucrose | 68.3 | ± | 4.5 | 55.2 | ± | 3.2 | 13.4 | ± | 1.7 | 59.4 | ± | 3.3 | 28.0 | ± | 1.4 | 27.9 | ± | 1.3 |
| Erythritol | 8.1 | ± | 3.1 | 8.7 | ± | 3.4 | 28.7 | ± | 3.2 | NQ[d] | | | NQ[d] | | | NQ[d] | | |
| Mannitol | 0.1 | ± | 0.01 | 0.2 | ± | 0.01 | <0.001 | | | <0.001 | | | 0.2 | ± | 0.01 | 0.8 | ± | 0.1 |
| Rhamnose | 0.1 | ± | 0.01 | 0.1 | ± | 0.01 | <0.001 | | | <0.001 | | | <0.001 | | | <0.001 | | |
| Arabinose | 0.3 | ± | 0.03 | 0.5 | ± | 0.1 | 0.9 | ± | 0.2 | 0.3 | ± | 0.03 | 1.2 | ± | 0.1 | 2.3 | ± | 0.2 |
| **Inorganic ions (µg mg$^{-1}$)** | | | | | | | | | | | | | | | | | | |
| Sodium | 0.25 | ± | 0.20 | 0.23 | ± | 0.01 | 0.30 | ± | 0.10 | 0.03 | ± | 0.002 | | | | | | |
| Ammonium | 1.36 | ± | 0.11 | 1.11 | ± | 0.90 | 0.89 | ± | 0.16 | 1.33 | ± | 0.13 | | | | | | |
| Potassium | 7.56 | ± | 0.81 | 6.43 | ± | 0.51 | 11.97 | ± | 0.16 | 5.22 | ± | 0.48 | | | | | | |
| Magnesium | 0.03 | ± | 0.00 | 0.05 | ± | 0.01 | 0.88 | ± | 0.01 | 0.74 | ± | 0.06 | NA[e] | | | | | |
| Calcium | 0.07 | ± | 0.02 | 0.12 | ± | 0.01 | 0.37 | ± | 0.03 | 1.85 | ± | 0.12 | | | | | | |
| Chloride | 0.40 | ± | 0.08 | 0.42 | ± | 0.05 | 2.10 | ± | 0.11 | 1.64 | ± | 0.18 | | | | | | |
| Nitrate | 0.19 | ± | 0.04 | 0.31 | ± | 0.11 | <0.019 | | | <0.019 | | | | | | | | |
| Phosphate | 3.94 | ± | 0.39 | 1.65 | ± | 0.41 | 10.5 | ± | 0.88 | 8.99 | ± | 0.87 | | | | | | |
| Sulfate | 0.79 | ± | 0.29 | 0.46 | ± | 0.02 | 0.94 | ± | 0.12 | 0.25 | ± | 0.03 | | | | | | |

[a] Purchased from Ploysciences

[b] Collected locally from Iowa City during late summer 2015

[c] Approximate diameters

[d] Not quantified (NQ) due to chromatographic interferences

[e] Not analyzed (NA)

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

# Figure captions

Figure 1: Temporal variation in precipitation and average temperature (a) in Iowa City, IA in the spring of 2013. Ambient concentrations of PM mass (b), glucose (c), fructose (d) and sucrose (e) in coarse and fine size fractions. The percent of PM and bioaerosol tracer mass in fine particles is shown on the right-axis for samples in which the analyte was detected in both size modes. During rain on 2 May, PM is suppressed, while pollen tracers in the fine mode substantially increased.

Figure 2: Temporal variation in precipitation and average temperature (a) in Iowa City, IA in the late summer of 2013. Ambient concentrations of PM mass (b), glucose (c), fructose (d) and sucrose (e) in coarse and fine size fractions. The percent of PM and bioaerosol tracer mass in fine particles is shown on the right-axis for samples in which the analyte was detected in both size modes. Fungal spore tracers increased significantly in the fine mode during the 2 May rain event.

Figure 3: Ambient concentrations of mannitol (a), glucans (b), and endotoxins (c) in coarse and fine size fractions in Iowa City, IA during spring of 2013. The percent of PM and bioaerosol tracer mass in fine particles is shown on the right-axis for samples in which the analyte was detected in both size modes. Fungal spore tracers increased significantly on 5 May, following a rainy period.

Figure 4: Ambient concentrations of mannitol (a), glucans (b), and endotoxins (c) in coarse and fine size fractions in Iowa City, IA during late summer of 2013. The percent of PM and bioaerosol tracer mass in fine particles is shown on the right-axis for samples in which the analyte was detected in both size modes Mannitol, the chemical tracer for fungal spores, and endotoxins from Gram-negative bacteria in fine mode increased on 22 August when it rained.

Figure 5: Apportionment of $PM_{10}$ mass (a) and $PM_{2.5}$ mass (b) during the spring of 2013 to pollens and fungal spores using CMB modeling.

Figure 6: Distribution of pollen and fungal spore mass (apportioned by the CMB model) across fine and coarse PM during dry and rainy conditions. The fine and coarse mode distributions of pollens and fungal spores shifted towards fine particles during rain, with a more pronounced effect for pollens compared to fungal spores.

## Figure 1

## Spring

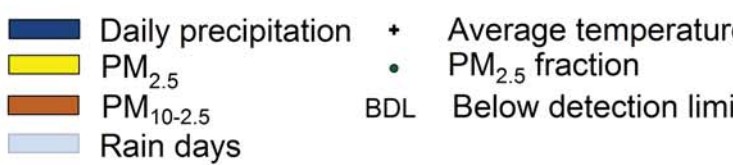

**(a)** Precipitation (mm) / Temperature (°C)

**(b)** PM mass (μg m⁻³) / Fraction in PM₂.₅ (%)

**(c)** Glucose (ng m⁻³)

**(d)** Fructose (ng m⁻³)

**(e)** Sucrose (ng m⁻³)

Apr 17, Apr 18, Apr 19, Apr 20, Apr 21, Apr 22, Apr 23, Apr 24, Apr 25, Apr 26, Apr 27, Apr 28, Apr 29, Apr 30, May 1, May 2, May 3, May 4, May 5, May 6, May 7, May 8, May 9

Daily precipitation
PM₂.₅
PM₁₀₋₂.₅
Rain days

+ Average temperature
● PM₂.₅ fraction
BDL  Below detection limit

# Figure 2

## Late summer

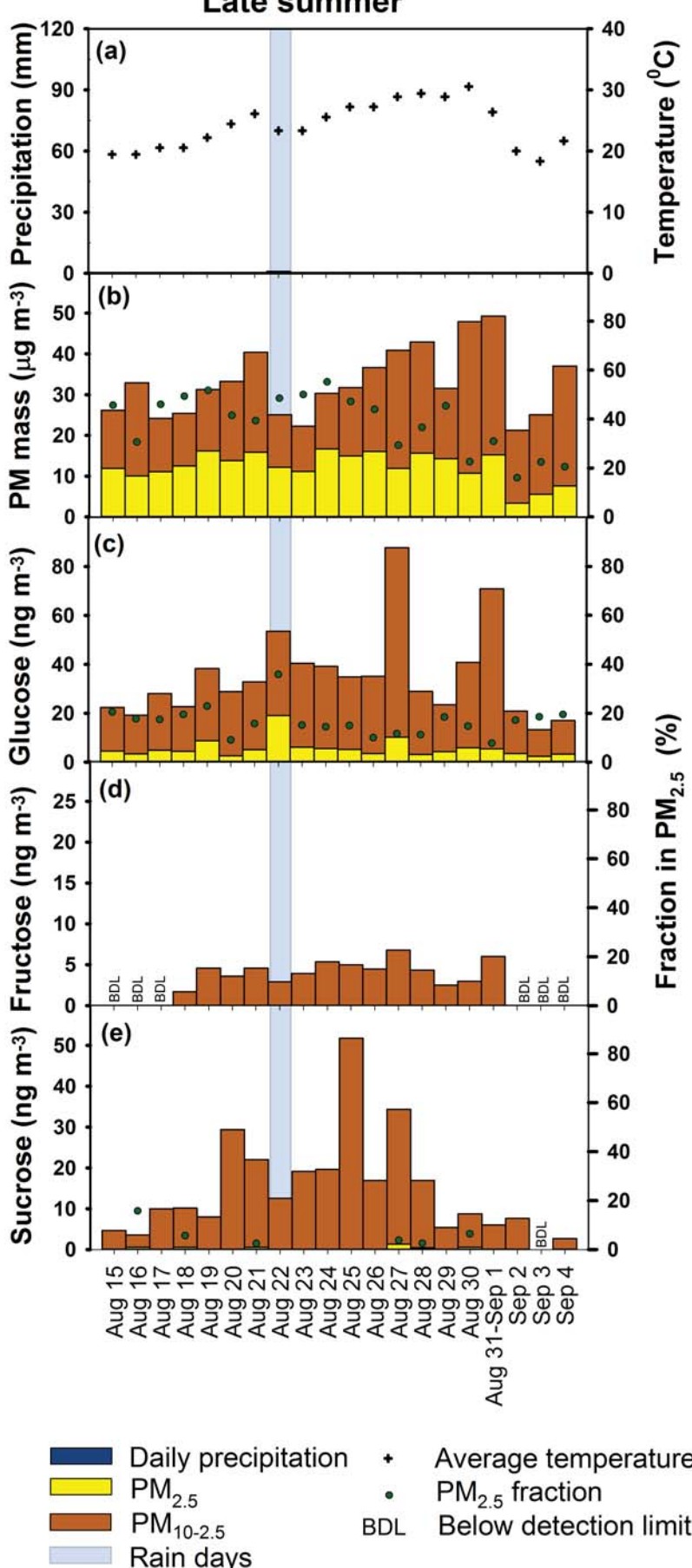

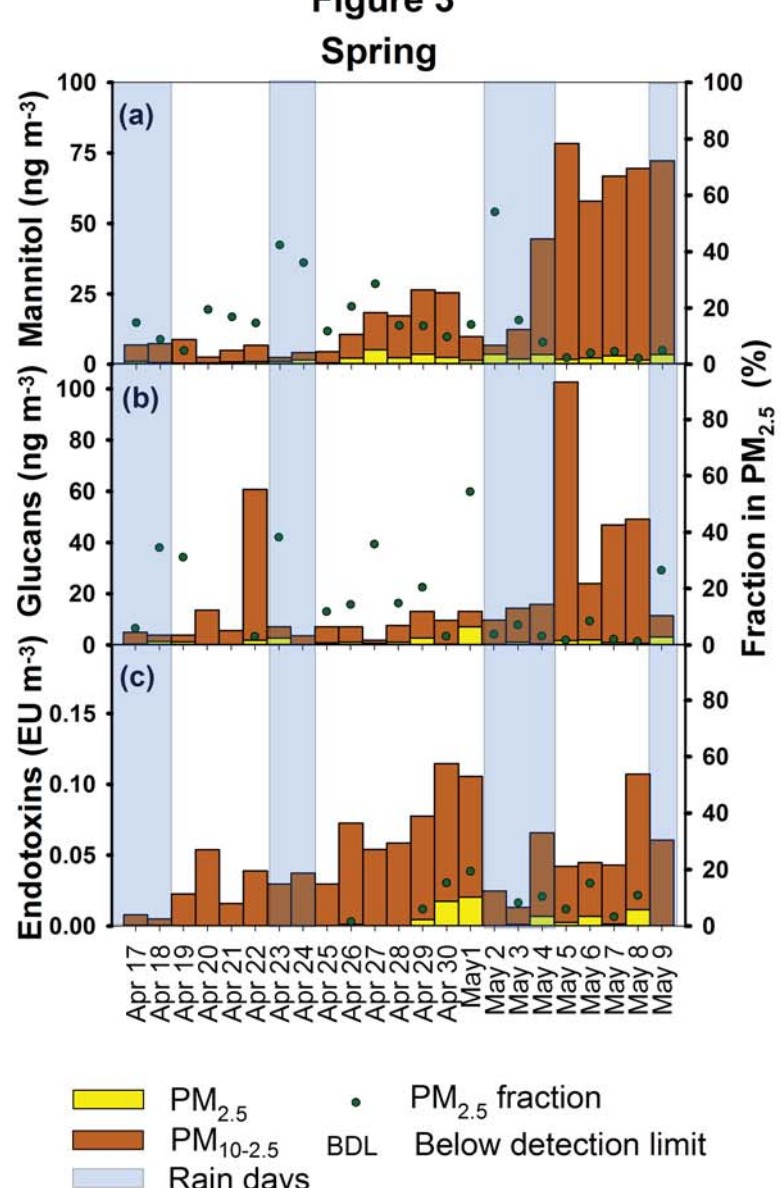

**Figure 3**

# Figure 4

## Late summer

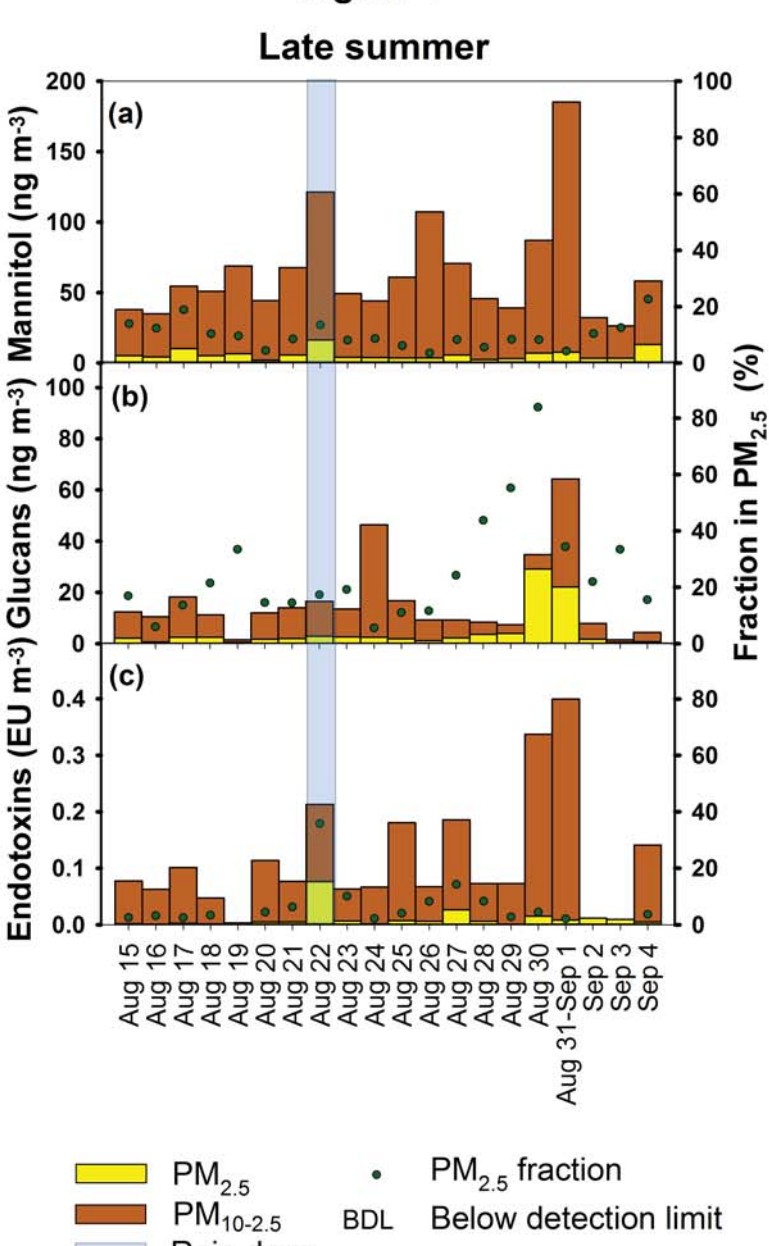

**Legend:**
- PM₂.₅
- PM₁₀₋₂.₅
- Rain days
- PM₂.₅ fraction
- BDL  Below detection limit

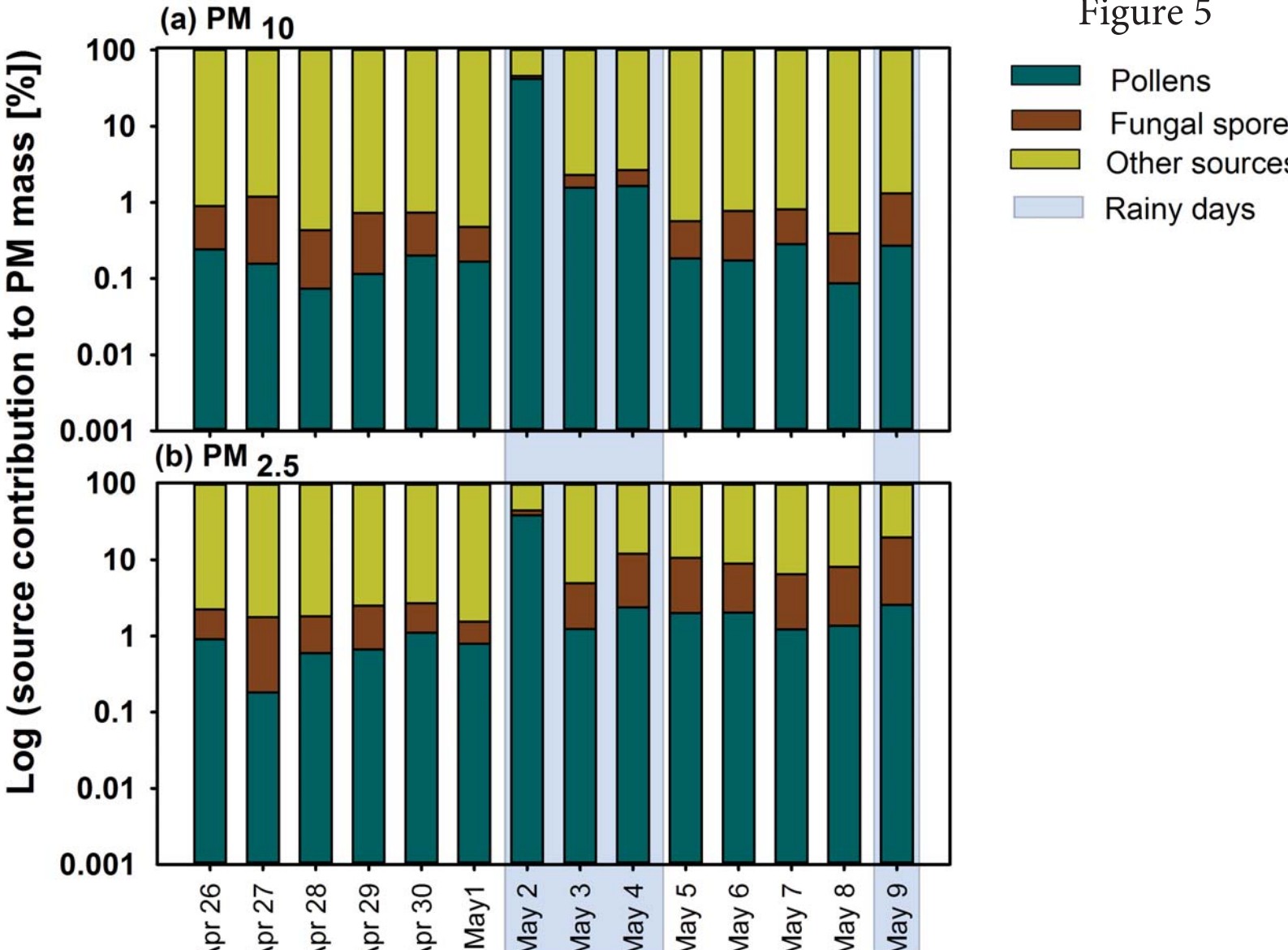

Figure 5

**Figure 6**

(a) Dry conditions — Pollens, Fungal spores

(b) Rainy days — Pollens, Fungal spores

Fine (<2.5 μm)
Coarse (2.5-10 μm)