# Peer review of "Influence of Rain on the Abundance of Bioaerosols in Fine and Coarse Particles"

_Atmospheric Chemistry and Physics, 2016_

## Referee Comment (RC1) · C. Morris (Referee) · 23 Oct 2016

GENERAL REMARKS

The overall objective of this work is to assess if rainfall influences the size distribution of biological aerosols and to identify the components of the aerosols – fungal, bacterial or pollen in particular – that contribute to the different size fractions. This question is important because fine aerosol particles move deeper into the respiratory tract thereby more readily setting off allergic reactions and allergies. For this work they have used chemical proxies for fungi, bacteria and pollen based on previously published reports and on additional work on chemical proxies of pollen as reported here.

The paper is well-written and the results are clear overall. Nevertheless, I have some questions and criticisms about the interpretation of their data and about the novelty of

their findings that need to be addressed. The specific questions are indicated below. More generally, as a biologist it is difficult to accept that only data about chemical proxies are sufficient for making specific conclusions about the presence, abundance and behavior of bacteria, fungi and pollen. I understand that chemical proxies are used because the nature of filters used for PM measurements are not compatible with microscopy. Furthermore, chemical analyses are more rapid and likely are more sensitive in terms of detection thresholds. But they are not as specific as needed for the many of the conclusions that the authors have made. In many of the studies where these chemical proxies were developed, other types of samplers were used in parallel to validate the results via microscopy. Pollen grains are rather large and have distinguishing features that can be recognized to aide in their identification and to differentiate them from certain fungal spores. The authors also report that pollen grains burst – because of the chemical signals they observed – without ever showing any direct evidence of this phenomenon, something that is also readily visible. Bauer et al 2003 (cited in the manuscript), noted that the relationship (regression coefficient) between the number of fungi in atmospheric samples and the quantities of the chemical proxies varied among different sampling sites and dates. This is likely because of the physiological changes that can occur throughout the life of fungi and especially in the production of different types of spores (ascopsores and conidia for ascomycetes; basiodiospore, picniospores, urediospores, aceiospores and teliospores for basidiomycetes, for example). Among their various conclusions, the authors stated that sources other than fungi were responsible for the glucan detected in the samples for cases where glucans and mannitol were not correlated. These are the types of conclusions that should be verified with other data – either direct observations, plating on growth media or through DNA analyses.

My second general question concerns the originality of the conclusions about how rainfall enhances the relative abundance of small aerosol particles as compared to larger particles. There is a growing body of literature describing how rainfall scavenges aerosol particles depending on their size – that have not been cited in this manuscript.

I have indicated some of those papers below in the specific comments section. The authors state that the medical community is well aware of the increase in cases of asthma after thunderstorms. If the authors have presented new information in understanding this phenomenon, then they should better acknowledge that in the paper. As a last comment, I am not sure why the authors mention CCN, IN and cloud processes in the manuscript. This manuscript concerns bioaerosols that impact human health. Mentioning CCN and IN does not add anything to the manuscript and it distracts a bit from the main message.

SPECIFIC REMARKS

Pg 2, Ln 16 : There is probably better terminology than "growing cycle". "Plant phenology" would be more appropriate.

Pg 3, Ln 1. What do the authors mean by "Bacteria in the atmosphere are typically settled on soil or vegetative surfaces" ?

Pg 3, Ln 3-6 : The authors state : "In vegetation covered areas, atmospheric bacterial concentrations peaked after approximately 1 h of rain relative to areas with bare soil (Robertson and Alexander, 1994)." However, this statement is not supported by this paper. Roberston and Alexander studied one single bacterial species (a nitrogen fixer that nodulates stems) and rainfall was simulated in their study. So it is not appropriate to make such generalizations from this one work.

Pg 3, Ln 9-10 : In support of the statement "bioaerosols in the atmosphere promote cloud and ice nucleation" the authors cite Pope, 2010; Sun and Ariya, 2006; Franc and Demott, 1998. However, these papers concern CCN and do not support the statement about ice nucleation. Please add a reference about ice nucleation if you are going to maintain information in the introduction and discussion about cloud physical processes. But as noted above in the general remarks, the focus of this work seems to be on aerosols that affect human health. The statements about aerosols that influence cloud processes seem irrelevant to the point of this research.

Pg 3, Ln 31: The authors do not state objectives that specifically mention the role of rain or the response of bioaerosols to rain. Why not?

Pg 4, Ln 9. In the methods section the authors do not indicate where the Andersen sampler is positioned relative to the ground and surrounding objects. How high above the ground was the Andersen sampler placed? What was the surrounding area like? Where there hedges, etc. Can the authors describe the footprint? the fetch? How was the sampler protected from rain? Did air circulate freely around the sampler? The authors need to provide information so that the reader can assess the representativeness of the air sampler relative to the surroundings.

Pg 6, the section starting on Ln 9: What was the purpose of the microscopy? How was this used in the study? Furthermore, why do the authors show a few images of pollen grains as one of the figures?

Pg 7, Ln 22: change "Rainfall corresponding to low PM" to "Rainfall corresponded to . . ."

Pg 7, Ln 26-28 : The authors state that "The shift in the PM size distribution of PM reflects that rain was more effective at scavenging and/or suppressing the release of coarse particles compared to fine particles." This is what should be expected. They should cite the relevant references here and in their discussion. The differential effect of scavenging according to particle size has been reported as early as the 1960's in the work of Gregory [Gregory, P. H. 1961. The Microbiology of the Atmosphere. New York: Interscience Publishers, Inc.]. For a more recent example, the authors should refer to [Li et al. 2016. Observed changes in aerosol physical and optical properties before and after precipitation events. Advances in Atmospheric Sciences 33: 931–944].

Pg 7, Ln 32: Change "levels are shown in Figure 3b" to "levels as shown. . .."

Pg 9, Ln 31-32 : The authors state that "Rain influenced ambient concentrations and the size distributions of fungal spore tracers, by triggering passive and active release

mechanisms." This is a very strong statement about mechanisms that is not supported by any biological observations in this work. This is a possible mechanism and it should be stated as a conjecture. Are there any other possible explanations such as growth, breaking of fungal hyphae, etc ?

Pg 9, Ln 34: The authors wrote: "Known for releasing spores after rain are some Ascospores....". "Ascospores" is not the correct terminology here. Ascospores are a type of spore. Here you mean Ascomycetes, i.e. a name for the group of fungi that produces ascospores during their sexual stage of reproduction. But although Ascomycetes are abundant, many of them produce mostly conidia that are formed on fungal "stems" called conidiophores and do not involve the formation of asci (sacs) containing ascospores and the accompanying fluids that are released into the atmosphere upon ascopore ejection. The relative prevalence of different types of spores (ascospores vs. conidia for the Ascomycetes and basidiospores vs. picnia, aeciospores and urediospores for Basidiomycetes) could be part of the reason that Bauer et al 2003 observed different relationships between the amount of chemical proxy and amount of atmospheric fungi depending on site and season.

Pg 10, Ln 23-24: The authors state: "and prior observations that pathogenic bacteria that grow on crops (i.e. Agrobacterium spp., and Rhizobium spp.) contain glucans in their structure". In this section the authors are trying to provide information about sources other than fungi for glucans in the atmosphere. Glucans are widely distributed in the microbial world and in biology in general. Here they give an example of 2 bacterial species. Although the information is accurate that these species contain glucans, they are soil-borne microorganisms. Furthermore, Rhizobium is not a pathogen, but rather it is a symbiotic nitrogen-fixing bacterium that is considered to be very beneficial to plants (NB: being beneficial or not has nothing to do with the likelihood of being airborne. I mention this only to clarify that it is not a pathogen). It is not logically obvious that these soil-borne bacterial species would be readily in the air. There have been reports of aerial dissemination of Rhizobium between African and the Canary Islands,

but this is also associated with loss of soils. It would be more appropriate to find a reference for the presence of glucans in bacteria in general, or to find references about bacteria that are common on aerial plant surfaces and more likely to be regularly in the atmosphere in agricultural contexts.

Pg 10, Ln 24-25: The authors state: "Agricultural crops are abundant in Iowa during the growing season and the mechanical agitation of plant surfaces by wind can aerosolize surface bacteria". Perhaps this is just awkward phrasing, but it should be changed because it suggests that the authors do not know that this is common knowledge. The "growing season" generally means the season during which crops grow. If Iowa were covered by forests, one would talk about the seasons (spring, summer, etc.). So, saying that agricultural crops are abundant during the growing season is redundant. Furthermore, I think that it is common knowledge that the Midwestern states of the US such as Iowa, Nebraska, Kansas, etc. are mostly covered by agriculture (corn, wheat, alfalfa). In this context, this sentence is surprising. It is sort of like reminding us, for example, that China or India have large populations of people.

Pg 13, Ln 13: The information on CCN and IN seems out of place in this paper because the authors are focusing on impacts on human health. For more detailed information about the possible sources of bioaerosols during and after rainfalls, I suggest that the authors refer to: Morris et al 2016 (http://journals.ametsoc.org/doi/abs/10.1175/BAMS-D-15-00293.1).

Pg 13, Ln 22-23: The authors state: "Elevating ambient fungal spore levels, particularly from species like Ascospores and Cladosporium, trigger allergenic respiratory diseases . . ." Here again, note that "Ascospores" is not a species. You cannot replace it with "Ascomycetes" because this is the name given to the members of the phylum Ascomycota. Perhaps you meant Aspergillus?

Pg 13, Ln 32, the authors describe the well-known phenomenon of thunderstorm asthma where allergies increase because of the abundance, after a storm, of small

particles that penetrate deep into the respiratory system. In light of the previous research on this phenomenon, the originality of this present work is not clear. They authors should point out more strongly how the work presented in this manuscript goes beyond what was currently known.

Pg 14, Ln 18-19: The authors state: "Warmer temperatures promoted pollen, fungal and bacterial growth leading to higher ambient levels of these bioaerosols during both spring and late summer periods." They state this in the Conclusion section as if they had observed this in this work. But isn't this what they infer from their observations of chemistry ? It would be more appropriate to say that the warm temperatures promoted increases in the proxies that are assumed to represent these organisms.

Pg 14, Ln 35-36: The authors state "The fragmentation of pollens due to osmotic rupture, shown previously only through microscopy methods, is demonstrated in this study for the first time by way of chemical tracers." However, in this current work they have not made any microscopic observations to verify the phenomenon of fragmentation. Without direct observation they cannot make this conclusion. They can only speculate.

---

## Referee Comment (RC2) · Anonymous Referee #2 · 5 Nov 2016

General Remarks:

In this study, the abundance of different bioaerosols (pollens, fungal spores and bacteria) present in both fine and coarse fraction of atmospheric aerosols is measured using chemical tracer method. The changes in the ambient concentration of bioaerosols and their relative abundance in different size fraction in response to variation in environmental conditions, especially rainfall were also assessed. Additionally the authors have also characterized the chemical profiles of different regionally abundant pollens and have estimated the pollen and fungal spore contribution to PM mass by using CMB modeling.

On general reading, the findings reported in the paper are quite interesting, however they are inconsistent in certain places. In this study since the authors quantify the

atmospheric abundance of different bioaerosols in only two broad size ranges (PM2.5 and PM2.5-10), I feel the use of term "size distribution" is inappropriate and misleading. In addition to chemical tracer analysis the authors have not given any other supporting results to further strengthen their finding of presence of smaller fraction of bioaerosols during the rain events.

Specific Remarks:

Fig.1, shows the microscopic images of pollen which are >20 $\mu$m. How is it relevant to show these images here as the authors are not measuring PM > 10 $\mu$m. Also these are not the images of pollens being measured from ambient atmosphere during any of the mentioned measurement periods. Instead of these images it make more sense to show images of ruptured pollens either collected from ambient atmosphere or from laboratory studies, which could further support their argument of presence of pollen fragments < 2.5 $\mu$m in size.

In Fig. 2, PM2.5-10 mass on April 17 and 18 appears to be zero. But there is glucose detected in this size fraction (Fig. 2c). How is this possible?

Page 5, L 27: Correct as Biomarkers.

Page 7, L 25: "Rain also affected the distribution of particles between the fine and coarse modes, with 48$\pm$11 % of PM10 was less than 2.5 $\mu$m on rainy days compared to 80$\pm$13 % on dry days". This sentence is confusing. Is the author mentioning about %contribution of PM2.5 in PM10 during wet and dry days?

Page 8, L 23: "passive release of larger pollen particles ranging 2.5–10 $\mu$m during others". What could be these larger pollen particles released passively during dry days? The microscopic images shows only pollens > 20 $\mu$m.

Page 9, L 1: From the ratio of glucose and sucrose the authors have related the pollens present on May 9 to that of red oak. Is there any other evidence to support this? The authors have mentioned in section 3.1 that the carbohydrate distribution in pollens are

likely to differ with change in environmental factors. Hence it is difficult to relate the pollens to any particular type only based on carbohydrate ratio. Also the authors have not reported any such match in carbohydrate ratios on any other days or in coarse mode PM fraction.

Page 9, L 16: "shift in glucose size distribution to 34% in the fine mode". It is not actually the size distribution, instead the relative contribution of glucose in fine fraction increases.

Page 9, L 23-24: "Daily concentrations of coarse mode concentrations of two fungal spore tracers—fungal sugar mannitol and the fungal cell wall component glucan—were significantly correlated with daily average temperature (rs>0.4, p<0.05)". This statement appears to be significant only for mannitol and not glucan. In L 27, The authors have reported an increase in fungal spore tracer level with increase in temperature. But no significant increase in glucan level can be seen in the graph (Fig. 4b).

Page10 L 1: "Fungal spore tracer levels dropped on days when rain fell (e.g. 23 April, 2 May), due to particle removal by wet deposition". The drop in tracer levels is visible only in coarse PM. The mannitol concentration in fine fraction actually shows an increase on 23 April as compared to the previous day without rain. Same is for glucan on 23 April. Also the relative contribution of fungal tracer in fine PM is high on these rainy days as compared to other dry days. Hence this statement is true only for tracer levels in coarse PM.

Page 10 L19: "which suggests an alternative non-fungal source of glucan". Pollen is a likely source'. If glucan can have other non-fungal source including pollen, then how can it be used as tracer for fungal spore. The high increase in glucan in PM10 on 22 April (Page 9, L 26) might be due to pollens which are generally released due during higher temperature.

Section 3.5: It is interesting to note that 92% of bacterial endotoxin is present in coarse

fraction of PM. Generally one would expect bacterial endotoxin to be abundant in fine fraction of PM as bacteria are smaller in size. Authors have not given any satisfactory explanation for this low concentration of endotoxin in fine PM.

Page 13, L 21-22: "The release of pollens, fungal spores, and Gram negative bacteria in fine particles during rain events, as observed surrounding spring and late-summer rain events in Iowa, has the potential to influence human health". This statement cannot be generalized at least for gram-negative bacteria during spring season where no increase in bacterial endotoxin in fine PM was observed during rain event.

Page 14, L 21-22: "Airborne fungal spore tracers, however, were suppressed by spring rain and increased in concentration following rain events". I feel this line contradicts the statement given in page 13, L 21, 'The release of pollens, fungal spores, and Gram negative bacteria in fine particles during rain events, as observed surrounding spring and late-summer rain events in Iowa, has the potential to influence human health".

---

## Author Comment (AC1) · 17 Dec 2016

Reviewer #1 (C. Morris) comments Reviewer 1 general comment: "The overall objective of this work is to assess if rainfall influences the size distribution of biological aerosols and to identify the components of the aerosols – fungal, bacterial or pollen in particular – that contribute to the different size fractions. This question is important because fine aerosol particles move deeper into the respiratory tract thereby more readily setting off allergic reactions and allergies. For this work they have used chemical proxies for fungi, bacteria and pollen based on previously published reports and on additional work on chemical proxies of pollen as reported here."

The paper is well-written and the results are clear overall. Nevertheless, I have some questions and criticisms about the interpretation of their data and about the novelty of

their findings that need to be addressed. The specific questions are indicated below. More generally, as a biologist it is difficult to accept that only data about chemical proxies are sufficient for making specific conclusions about the presence, abundance and behavior of bacteria, fungi and pollen. I understand that chemical proxies are used because the nature of filters used for PM measurements are not compatible with microscopy. Furthermore, chemical analyses are more rapid and likely are more sensitive in terms of detection thresholds. But they are not as specific as needed for the many of the conclusions that the authors have made. In many of the studies where these chemical proxies were developed, other types of samplers were used in parallel to validate the results via microscopy. Pollen grains are rather large and have distinguishing features that can be recognized to aide in their identification and to differentiate them from certain fungal spores. The authors also report that pollen grains burst – because of the chemical signals they observed – without ever showing any direct evidence of this phenomenon, something that is also readily visible. Bauer et al 2003 (cited in the manuscript), noted that the relationship (regression coefficient) between the number of fungi in atmospheric samples and the quantities of the chemical proxies varied among different sampling sites and dates. This is likely because of the physiological changes that can occur throughout the life of fungi and especially in the production of different types of spores (ascospores and conidia for ascomycetes; basiodiospore, picniospores, urediospores, aceiospores and teliospores for basidiomycetes, for example). Among their various conclusions, the authors stated that sources other than fungi were responsible for the glucan detected in the samples for cases where glucans and mannitol were not correlated. These are the types of conclusions that should be verified with other data – either direct observations, plating on growth media or through DNA analyses.

My second general question concerns the originality of the conclusions about how rainfall enhances the relative abundance of small aerosol particles as compared to larger particles. There is a growing body of literature describing how rainfall scavenges aerosol particles depending on their size – that have not been cited in this manuscript.

I have indicated some of those papers below in the specific comments section. The authors state that the medical community is well aware of the increase in cases of asthma after thunderstorms. If the authors have presented new information in understanding this phenomenon, then they should better acknowledge that in the paper. As a last comment, I am not sure why the authors mention CCN, IN and cloud processes in the manuscript. This manuscript concerns bioaerosols that impact human health. Mentioning CCN and IN does not add anything to the manuscript and it distracts a bit from the main message.

Response to reviewer #1 general comment: We thank the reviewer for their input and detailed comments that bring a valuable biological perspective on this data set. We have revised the manuscript in response to each specific comment point-by-point below. We summarize our responses to the main concerns of the reviewer here:

With respect to our use of chemical proxies to study bioaerosols, we note that this approach has been taken previously. We consider a strength of this work to be the combination of chemical tracers and biological assays, which have been combined in only a few prior studies (Rathnayake et al., 2016; Chow et al., 2015). We agree with the reviewer that these methods have limitations, particularly in the ability to identify bioaerosols at the species level, and have clarified this in the manuscript by adding the following text at page 7 line 5-14: "Measurements of chemical tracers and biological markers are used to determine the relative concentrations and distribution of pollens, fungal spores, and bacteria in fine and coarse PM. Only few prior studies have combined chemical tracers and biological markers (Rathnayake et al., 2016; Chow et al., 2015), while many others have relied on either chemical tracers (Fu et al., 2012; Medeiros et al., 2006; Burshtein et al., 2011; Yttri et al., 2007; Zhang et al., 2010) or biological assays (Nilsson et al., 2011; Mueller-Anneling et al., 2004; Pavilonis et al., 2013; Madsen et al., 2011; Singh et al., 2011). Glucose, fructose, and sucrose are major components of pollens, mannitol and fungal glucans are in fungal spores, and endotoxins are in bacteria. In the ambient particulate matter, these species are

used as bioaerosol tracers, since their concentrations reflect mass concentrations of the corresponding bioaerosol. These species provide general insight to classes of bioaerosols present, but cannot be used for species-level identification, which requires either microscopy imaging or DNA sequencing."

We agree with the reviewer that additional, corroborating measurements of bursting pollens in the PM samples collected would be very useful; however the PM samples were collected on filters that were not conducive to microscopy analysis and the chemical tracer analysis and biological assays were destructive, so additional measurements, such as microscopy or DNA sequencing were not possible. The study of intact pollens by chemical methods links the chemical tracers to local pollen types (namely oak) and the scientific literature base provides evidence of pollen rupturing that we draw upon in discussing our results. In future studies of bioaerosols, we plan to incorporate additional analytical tools, as suggested by the reviewer.

In regards to the reviewer's comment on glucan and mannitol correlations, we have re-worded our sentences as described in reviewer 1 specific remarks 18 and 19. To incorporate reviewer 1 comments on fungal spore tracer ratios of different types of fungal spores as a likely reasoning for the lack of correlation of mannitol and glucans we revised our discussion at page 11, lines 8-12 as described in reviewer 1 specific remark 12.

In regards to the reviewer's second general comment, we have made a number of modifications to the text. To address the comment about rain suppressing atmospheric PM, we have added the suggested references and expanded the discussion as suggested by the reviewer in response to reviewer 1 specific remark 9. In order to acknowledge the novelty of this study and new insights to thunderstorm asthma, we incorporated a paragraph to the manuscript as described in response to reviewer 1, specific remark 17. We agree with the reviewer that the main implication of the observations in this study relate to human health and asthma, although the release of bioaerosols to fine PM has also important implications for meteorology as they can be effective cloud condensation nuclei (CCN) and ice nuclei (IN). We believe that this is important to include, albeit briefly, as noted in response to reviewer 1's specific remarks 4 and 15. The changes made to this manuscript in response to these suggestions are detailed below.

Reviewer #1 specific remark 1: "Pg 2, Ln 16 : There is probably better terminology than "growing cycle". "Plant phenology" would be more appropriate."

Response to reviewer #1 specific remark 1: We agree with the reviewer and changed the wording in page 2, line 16. Now the text in page 2, line 16 reads as "Ambient levels of pollens vary seasonally with plant phenology (Galán et al., 1995; Targonski et al., 1995)."

Reviewer #1 specific remark 2: "Pg 3, Ln 1. What do the authors mean by "Bacteria in the atmosphere are typically settled on soil or vegetative surfaces" ?

Response to reviewer #1 specific remark 2: We thank the reviewer for pointing this out. In order to make our statement more clear we changed the wording in page 3, line 1. Now page 3 line 1 reads as "Bacteria in the atmosphere are typically attached to soil or vegetative surfaces as agglomerations of cells (Jones and Harrison, 2004)."

Reviewer #1 specific remark 3: "Pg 3, Ln 3-6 : The authors state: "In vegetation covered areas, atmospheric bacterial concentrations peaked after approximately 1 h of rain relative to areas with bare soil (Robertson and Alexander, 1994)." However, this statement is not supported by this paper. Roberston and Alexander studied one single bacterial species (a nitrogen fixer that nodulates stems) and rainfall was simulated in their study. So it is not appropriate to make such generalizations from this one work.

Response to reviewer #1 specific remark 3: We appreciate the reviewer pointing this out and have added additional information and citations to support a more general statement. The text at page 3, line 4-6 now reads as "In vegetation covered areas, atmospheric bacterial concentrations have been shown to increase during and after simulated rain events (Graham et al., 1977; Robertson and Alexander, 1994) as well

as natural rain events (Constantinidou et al., 1990; Huffman et al., 2013)."

Reviewer #1 specific remark 4: "Pg 3, Ln 9-10 : In support of the statement "bioaerosols in the atmosphere promote cloud and ice nucleation" the authors cite Pope, 2010; Sun and Ariya, 2006; Franc and Demott, 1998. However, these papers concern CCN and do not support the statement about ice nucleation. Please add a reference about ice nucleation if you are going to maintain information in the introduction and discussion about cloud physical processes. But as noted above in the general remarks, the focus of this work seems to be on aerosols that affect human health. The statements about aerosols that influence cloud processes seem irrelevant to the point of this research.

Response to reviewer #1 specific remark 4: We agree with the reviewer that the current set of references only supports the CCN activity of bioaerosols. CCN and IN are a very active research field, although of secondary importance to health, the results of this study suggest that the pollen bursting phenomenon would impact CCN and IN levels. Therefore in response to this comment, we expanded the reference list to include references that showed IN activity of bioaerosols. Now page 3, line 9 reads as "Once released, bioaerosols in the atmosphere promote cloud and ice nucleation (Pope, 2010; Sun and Ariya, 2006; Murray et al., 2012)"

Reviewer #1 specific remark 5: "Pg 3, Ln 31: The authors do not state objectives that specifically mention the role of rain or the response of bioaerosols to rain. Why not?

Response to reviewer #1 specific remark 5: We thank the reviewer for pointing this and in response we have revised our objectives to be more specific. Page 3, lines 31-33 now reads as "Our central objectives were. . . . . ..ii) evaluate environmental conditions including rain and temperature that lead to high levels and decreases in bioaerosol sizes across fine (PM2.5) and coarse (PM10-2.5) modes. . ...."

Reviewer #1 specific remark 6: "Pg 4, Ln 9. In the methods section the authors do not indicate where the Andersen sampler is positioned relative to the ground and surrounding objects. How high above the ground was the Andersen sampler placed? What was the surrounding area like? Where there hedges, etc. Can the authors describe the footprint? the fetch? How was the sampler protected from rain? Did air circulate freely around the sampler? The authors need to provide information so that the reader can assess the representativeness of the air sampler relative to the surroundings.

Response to reviewer #1 specific remark 6: As suggested by the reviewer, we have added details to the site and sampler descriptions at page 4, lines 8-22:"Daily (24 h) PM samples were collected from 17 April–9 May (springtime) and 15 August–04 September (late-summer) in 2013, at the University of Iowa air monitoring site in Iowa City, Iowa, US (+41.6647, − 91.5845). The site was located at the University of Iowa Practice Fields in a suburban landscape in an open area surrounded by woods, agricultural fields, meadows and a parking lot. PM2.5 and PM10-2.5 were collected using an Andersen dichotomous sampler (Series 241) that included a PM10 cutoff impactor (Anderson Instruments, Model 246b) and virtual impactor. The total air flow rate was 16.67 L min-1 and the coarse flow rate was 1.667 L min-1. PM samples were collected on 37-mm Teflon filters (Pall Corp.) and PM10 was determined as the sum of PM2.5 and PM10-2.5. The dichotomous sampler had a UMLBL (the University of Minnesota-Lawrence Berkeley Laboratory) type inlet which is equipped with a rain guard and a mesh-screen to exclude rain drops and insects. An additional set of PM2.5 samples were collected on to 90-mm quartz fibre filters (Pall Life Sciences) using a medium-volume sampler (URG Corp.) equipped with a sharp-cut cyclone to select PM2.5 at a flow rate of 90 L min-1. Rain was excluded from the PM2.5 sampler primarily by positioning the inlet downward and secondarily by the cyclone. Both samplers were affixed to a platform 3 m above ground level and were unobstructed. Flowrates were measured using a rotameter at the beginning and the end of each sampling period; average flowrates were used to calculate air volumes Filters were changed at 08:00 local time (CST) and one field blank was collected for every 5 samples. After sample collection, filters were stored at -20 ËŽC in the dark." Reviewer #1 specific remark 7: "Pg 6, the section starting on Ln 9: What was the purpose of the microscopy? How

was this used in the study? Furthermore, why do the authors show a few images of pollen grains as one of the figures?

Response to reviewer #1 specific remark 7: We agree with the reviewer the need to clarify the use of microscopy in the manuscript, which was specifically to determine the diameter of pollen grains. We expanded our objectives to include why we took microscopy measurements of pollens. In the introduction section page 3, line 31-34 reads as "Our central objectives were. . . . . . iii) determine intact pollen diameters and chemically profile regionally-important pollen types (red oak, pin oak, cotton ragweed, giant ragweed and corn) for use in source apportionment. . ."

Moreover we incorporated the purpose of doing microscopy measurements in the method, section 2.5. Now page 6, lines 16-18 reads as "Pollen images were taken to determine pollen grain diameters using a Zeiss LSM 710 fluorescence microscope (Carl Zeiss Microscopy GmbH, 07745 Jena, Germany) following PÓğhlker et al. (2012), and IX-81 inverted microscope (Olympus Corporation, Tokyo, Japan).

We also agree with the reviewer that these microscopy measurements were not used in this study other than to visualize the pollen size and shape thus we moved the images of pollens (Figure 1) to the supplementary information Figure S1."

Reviewer #1 specific remark 8: "Pg 7, Ln 22: change "Rainfall corresponding to low PM" to "Rainfall corresponded to : : : "

Response to reviewer #1 specific remark 8: We agree with the reviewer and text in page 7, line 22 is revised. Now page 8 line 2 reads as "Rainfall corresponded to low PM concentrations with average. . ."

Reviewer #1 specific remark 9: "Pg 7, Ln 26-28 : The authors state that "The shift in the PM size distribution of PM reflects that rain was more effective at scavenging and/or suppressing the release of coarse particles compared to fine particles." This is what should be expected. They should cite the relevant references here and in their

discussion. The differential effect of scavenging according to particle size has been reported as early as the 1960's in the work of Gregory [Gregory, P. H. 1961. The Microbiology of the Atmosphere. New York: Interscience Publishers, Inc.]. For a more recent example, the authors should refer to [Li et al. 2016. Observed changes in aerosol physical and optical properties before and after precipitation events. Advances in Atmospheric Sciences 33: 931–944].

Response to reviewer #1 specific remark 9: We agree with the reviewer's suggestion to provide citations in support of this statement. The revised text in page 8, lines 6-12 reads: "The shift in the PM size distribution reflects that rain was more effective at scavenging and/or suppressing the release of coarse particles compared to fine particles. This is consistent with previous ambient studies that have demonstrated coarse PM is more effectively scavenged than fine particles (Guo et al., 2016; Li et al., 2016). Particle removal via rainfall depends on many factors including a strong dependence on the particle size (Gregory, 1962; Baklanov and Sørensen, 2001); airborne particles with diameters greater than 3 $\mu$m have a higher tendency to collide with falling rain drops and are effectively scavenged via inertial impaction (Wang et al., 2010; Andronache, 2003; Mircea et al., 2000)."

Reviewer #1 specific remark 10: "Pg 7, Ln 32: Change "levels are shown in Figure 3b" to "levels as shown: : :."

Response to reviewer #1 specific remark 10: We agree with the reviewer and we revised the text in page 7, line 32 accordingly. Now page 8, line 16 read as ". . .levels as shown in Figure 2b. . ."

Reviewer #1 specific remark 11: "Pg 9, Ln 31-32 : The authors state that "Rain influenced ambient concentrations and the size distributions of fungal spore tracers, by triggering passive and active release mechanisms." This is a very strong statement about mechanisms that is not supported by any biological observations in this work. This is a possible mechanism and it should be stated as a conjecture. Are there any

other possible explanations such as growth, breaking of fungal hyphae, etc ?

Response to reviewer #1 specific remark 11: We appreciate this reviewer for highlighting this. We agree with the comment and we revised the text in page 9, line 31-32 accordingly. Now the revised text in page 10, lines 19-23 reads as "Rain influenced ambient concentrations and the size distributions of fungal spore tracers, likely by triggering passive and/or active release mechanisms and/or promoting fungal growth. Maximum mannitol and glucan levels occurred on 5 May, which followed three days with rain (Figure 3a-b). Rainfall facilitates fungal growth promoting fungal germination and hyphal growth (Schulthess and Faeth, 1998; Morris et al., 2016) and wet conditions that follow rain are favourable for active release of fungal spores (Rodriguez Rajo et al., 2005; Van Osdol et al., 2004)."

Reviewer #1 specific remark 12: "Pg 9, Ln 34: The authors wrote: "Known for releasing spores after rain are some Ascospores: : :.". "Ascospores" is not the correct terminology here. Ascospores are a type of spore. Here you mean Ascomycetes, i.e. a name for the group of fungi that produces ascospores during their sexual stage of reproduction. But although Ascomycetes are abundant, many of them produce mostly conidia that are formed on fungal "stems" called conidiophores and do not involve the formation of asci (sacs) containing ascospores and the accompanying fluids that are released into the atmosphere upon ascopore ejection. The relative prevalence of different types of spores (ascospores vs. conidia for the Ascomycetes and basidiospores vs. picnia, aeciospores and urediospores for Basidiomycetes) could be part of the reason that Bauer et al 2003 observed different relationships between the amount of chemical proxy and amount of atmospheric fungi depending on site and season.

Response to reviewer #1 specific remark 12: We agree with the reviewer and appreciate their explanation of fungal spore types in prominent fungal species. We revised the text in page 10, line 22-25 to read ". ....wet conditions that follow rain are favourable for active release of fungal spores (Rodriguez Rajo et al., 2005; Van Osdol et al., 2004). For instance, actively discharged ascospores peak after rain in wet conditions (Troutt

and Levetin, 2001; Elbert et al., 2007; MacHardy and Gadoury, 1986)."

To address the reviewer comment about the relative prevalence of different types of spores and chemical proxies (both here and in their general comments), we have incorporated the likelihood of different spore types into our discussion of fungal spore tracers. Now page 11, lines 8-12 reads as "Coarse mode glucan concentrations in late summer were neither correlated with temperature (rs=0.01, p=1), nor mannitol (rs=0.2, p=0.3). Mannitol concentrations and fungal spore counts have spatial and seasonal differences from one another (Bauer et al., 2008), likely due to differences in mannitol emission per spore across fungal types (Elbert et al., 2007; Bauer et al., 2008) and/or mannitol concentrations in spores from within a species (e.g. ascomycetes releases ascospores during sexual reproduction and conidia during asexual reproduction (Nauta and Hoekstra, 1992))."

Reviewer #1 specific remark 13: "Pg 10, Ln 23-24: The authors state: "and prior observations that pathogenic bacteria that grow on crops (i.e. Agrobacterium spp., and Rhizobium spp.) contain glucans in their structure". In this section the authors are trying to provide information about sources other than fungi for glucans in the atmosphere. Glucans are widely distributed in the microbial world and in biology in general. Here they give an example of 2 bacterial species. Although the information is accurate that these species contain glucans, they are soil-borne microorganisms. Furthermore, Rhizobium is not a pathogen, but rather it is a symbiotic nitrogen-fixing bacterium that is considered to be very beneficial to plants (NB: being beneficial or not has nothing to do with the likelihood of being airborne. I mention this only to clarify that it is not a pathogen). It is not logically obvious that these soil-borne bacterial species would be readily in the air. There have been reports of aerial dissemination of Rhizobium between African and the Canary Islands, but this is also associated with loss of soils. It would be more appropriate to find a reference for the presence of glucans in bacteria in general, or to find references about bacteria that are common on aerial plant surfaces and more likely to be regularly in the atmosphere in agricultural contexts.

[Figure]

Response to reviewer #1 specific remark 13: We agree with this reviewer comment and page 10, line 23-24 is revised to reflect the presence of glucans in bacteria in general. Page 11, line 18 now read as "Alternatively glucans may have derived from bacterial cells (McIntosh et al., 2005; Rylander and Lin, 2000), although their correlation was not significant (rs=0.4, p=0.1) ."

Reviewer #1 specific remark 14: "Pg 10, Ln 24-25: The authors state: "Agricultural crops are abundant in Iowa during the growing season and the mechanical agitation of plant surfaces by wind can aerosolize surface bacteria". Perhaps this is just awkward phrasing, but it should be changed because it suggests that the authors do not know that this is common knowledge. The "growing season" generally means the season during which crops grow. If Iowa were covered by forests, one would talk about the seasons (spring, summer, etc.). So, saying that agricultural crops are abundant during the growing season is redundant. Furthermore, I think that it is common knowledge that the Midwestern states of the US such as Iowa, Nebraska, Kansas, etc. are mostly covered by agriculture (corn, wheat, alfalfa). In this context, this sentence is surprising. It is sort of like reminding us, for example, that China or India have large populations of people.

Response to reviewer #1 specific remark 14: In response to reviewer 1, specific comment 13, we generalized our discussion, and the sentence in question has been deleted.

Reviewer #1 specific remark 15: "Pg 13, Ln 13: The information on CCN and IN seems out of place in this paper because the authors are focusing on impacts on human health. For more detailed information about the possible sources of bioaerosols during and after rainfalls, I suggest that the authors refer to: Morris et al 2016 (http://journals.ametsoc.org/doi/abs/10.1175/BAMSD- 15-00293.1).

Response to reviewer #1 specific remark 15: We think the discussion of CCN and IN activity of bioaerosols to be relevant to this work, particularly with respect to observa-

tions of pollen tracers in fine PM that are more CCN active than coarse PM. Consequently, we have retained this component of the manuscript. As suggested, we refer to Morris et al., 2016 about possible sources of bioaerosols during and after rain. The revised text follows.

In the revised manuscript, the text at page 14, lines 9-21 reads: "The release of fine sized bioaerosols can influence cloud formation, by acting as CCN and IN. Pollen fragments are effective CCN and IN (Pope, 2010; Diehl et al., 2001). During rain intact pollen particles can swell and rupture, producing hundreds of fine-sized pollen particles (D'Amato et al., 2007), significantly increasing the number of CCN and IN active particles in the atmosphere. Bacteria and fungal spores also active IN and CCN (Murray et al., 2012; Sun and Ariya, 2006; Hassett et al., 2015) .Bacterial strains with higher IN activity (mostly Gram-negative bacteria that habitat plant surfaces (Murray et al., 2012), such as Pseudomanas syringae) increase in population during rain (Hirano et al., 1996), which can substantially increase airborne IN (Morris et al., 2016) that can persist in the atmosphere for weeks following rain (Bigg et al., 2015). Rainfall in general favours fungal growth (Schulthess and Faeth, 1998; Morris et al., 2016) as well as passive and active release of spores (Rodriguez Rajo et al., 2005; Van Osdol et al., 2004; Allitt, 2000; Elbert et al., 2007; Huffman et al., 2013) thereby increasing CCN and IN active particles in the atmosphere. When decreased in size ($< 2.5 \mu$m), these bioaerosols are more effective IN (Murray et al., 2015; Huffman et al., 2013). Because smaller particles have longer atmospheric lifetimes, fine bioaerosols will be transported longer distances before deposition, and thus may have effects in areas downwind of their release."

We also revised text in page 12, lines 9-23 to incorporate information from suggested references: "On 22 August, the only late summer day with rain, fine mode endotoxin concentrations reached a maximum (Figure 5c). Meanwhile, the endotoxin fraction in the fine mode increased to 36% relative to an average of 5% on dry days. Rainfall promotes bacterial growth, such as Pseudomanas syringae that are common on

plant surfaces and rapidly increase their populations during raining (Hirano and Upper, 1990; Hirano et al., 1996). The release of endotoxin to fine PM is expected to be caused by the aerosolization of Gram-negative bacteria living on plant surfaces (e.g., Pseudomanas syringae, Pseudomanas fluorescens, and Pseudomanas viridiflava etc. (Murray et al., 2012)) by agitation of plants or fungi by falling rain (Jones and Harrison, 2004; Constantinidou et al., 1990). Soil resuspension was suggested as an important source of bacterial endotoxins in spring (section 3.5.1), however coarse mode endotoxins were not significantly correlated with calcium in late summer (rs=0.2, p=0.33), suggesting that this is not the case. Consequently, non-soil bacterial sources were likely responsible, such as plant surfaces (Romantschuk, 1992; Jeter and Matthysse, 2005; Murray et al., 2012) that are probably agricultural row crops (Lindemann et al., 1982; Hirano et al., 1996) in the agricultural state of Iowa. This link could be further explored by examining the co-occurrence of bacterial endotoxins with markers of plant waxes (i.e. odd-numbered n-alkanes), but is beyond the scope of the present study. The comparison of spring and late-summer endotoxin behavior in response to rain suggests that soil bacteria are dominate in springtime, while bacteria residing on plant surfaces dominate in late-summer." The revisions done to the discussion of fungal spores are described in reviewer 1 specific remark 11.

Reviewer #1 specific remark 16: "Pg 13, Ln 22-23: The authors state: "Elevating ambient fungal spore levels, particularly from species like Ascospores and Cladosporium, trigger allergenic respiratory diseases : : :" Here again, note that "Ascospores" is not a species. You cannot replace it with "Ascomycetes" because this is the name given to the members of the phylum Ascomycota. Perhaps you meant Aspergillus?

Response to reviewer #1 specific remark 16: We agree with this reviewer comment and now page 14, lines 23-26 read as "Elevating ambient fungal spore levels, particularly from species like Penicillium, Aspergillus and Cladosporium, trigger allergenic respiratory diseases like allergic rhinitis and asthma (Garrett et al., 1998; Tillie-Leblond et al., 2011; Knutsen et al., 2012) and high environmental exposures may lead to asthma

exacerbations (Dales et al., 2003)."

Reviewer #1 specific remark 17: "Pg 13, Ln 32, the authors describe the well-known phenomenon of thunderstorm asthma where allergies increase because of the abundance, after a storm, of small particles that penetrate deep into the respiratory system. In light of the previous research on this phenomenon, the originality of this present work is not clear. They authors should point out more strongly how the work presented in this manuscript goes beyond what was currently known.

Response to reviewer #1 specific remark 17: To clarify the novelty of this work, we have added the following paragraph to the section 3.7 on page 15, lines 15-29: "The results of this study provide new insight and tools to better understand the potential scope of thunderstorm asthma. While thunderstorm asthma has been documented in a number of locations, the data presented herein provide the first evidence of this phenomenon occurring in the Midwestern US. Thunderstorms and heavy rain are common in this region during spring, and thus it is anticipated that conditions characteristic of thunderstorm asthma likely occur several times annually. Pollen prediction indices do not currently account for the release of fine pollen fragments during rain, and consequently sensitive populations are not forewarned. To understand the potential for conditions that trigger thunderstorm asthma more broadly, chemical tracer approaches, as used here, are a useful tool. Chemical tracers provide a sensitive method of detecting fine pollens particles that may be useful in monitoring conditions that precede PM2.5 pollen release. Because carbohydrates are not expected to undergo chemical alternation by the pollen bursting, they also provide a means of tracking pollens across PM size fractions and associating pollens with their species of origin. Microscopy-based methods are challenged by changes to particle size and morphology upon bursting, which may require use of multiple microscopy techniques suitable for different particle sizes. Chemical tracer methods have potential to be broadly applied, as national monitoring programs routinely collect PM2.5 samples on filters for chemical analysis. In this way, regions and atmospheric conditions that lead to high levels of PMǍň2.5 pollen particles

may be better defined." Reviewer #1 specific remark 18: "Pg 14, Ln 18-19: The authors state: "Warmer temperatures promoted pollen, fungal and bacterial growth leading to higher ambient levels of these bioaerosols during both spring and late summer periods." They state this in the Conclusion section as if they had observed this in this work. But isn't this what they infer from their observations of chemistry ? It would be more appropriate to say that the warm temperatures promoted increases in the proxies that are assumed to represent these organisms.

Response to reviewer #1 specific remark 18: We agree that this statement should be restated to align with the data we present. In response we edited page 13, line 18-19. Now page 16, lines 3-5 reads as "Elevated bioaerosol tracer levels were observed when temperatures are warmer suggesting increased pollen, fungal and bacterial concentrations during both spring and late summer periods."

Reviewer #1 specific remark 19: "Pg 14, Ln 35-36: The authors state "The fragmentation of pollens due to osmotic rupture, shown previously only through microscopy methods, is demonstrated in this study for the first time by way of chemical tracers." However, in this current work they have not made any microscopic observations to verify the phenomenon of fragmentation. Without direct observation they cannot make this conclusion. They can only speculate.

Response to reviewer #1 specific remark 19: We agree with the reviewer, in response, we re-worded the text in page 14, line 35-36. Now page 16, line 10-13 read as "
[revised manuscript text omitted]

---

## Author Comment (AC2) · 17 Dec 2016

Reviewer 2 general comments: "In this study, the abundance of different bioaerosols (pollens, fungal spores and bacteria) present in both fine and coarse fraction of atmospheric aerosols is measured using chemical tracer method. The changes in the ambient concentration of bioaerosols and their relative abundance in different size fraction in response to variation in environmental conditions, especially rainfall were also assessed. Additionally the authors have also characterized the chemical profiles of different regionally abundant pollens and have estimated the pollen and fungal spore contribution to PM mass by using CMB modeling."

On general reading, the findings reported in the paper are quite interesting, however they are inconsistent in certain places. In this study since the authors quantify the

atmospheric abundance of different bioaerosols in only two broad size ranges (PM2.5 and PM2.5-10), I feel the use of term "size distribution" is inappropriate and misleading. In addition to chemical tracer analysis the authors have not given any other supporting results to further strengthen their finding of presence of smaller fraction of bioaerosols during the rain events.

Response to reviewer #2 general comments: In response we changed the manuscript title from "Influence of Rain on the Abundance and Size Distribution of Bioaerosols" to "Influence of Rain on the Abundance of Bioaerosols in Fine and Coarse Particles".

The tracer analysis performed during this work was destructive, so additional filter-based measurements, such as microscopy studies were not possible after our analysis revealed high concentrations of pollen tracers in fine PM. We analyzed the diameter and carbohydrate profile of local pollens, which demonstrates that the chemical signature of oak pollens matches the fine PM carbohydrate profile, suggesting that oak pollens may have been the origin of fine pollen PM. Further confirmation by microscopy techniques would be useful, but are beyond the scope of this study, as samples were not collected on substrates conducive to microscopy analysis. We plan to incorporate microscopy into future studies of bioaerosols. In the absence of microscopy, we draw upon scientific literature that demonstrates the bursting of pollens under wet conditions.

Reviewer #2 specific remark 1: "Fig.1, shows the microscopic images of pollen which are >20 $\mu$m. How is it relevant to show these images here as the authors are not measuring PM > 10 $\mu$m. Also these are not the images of pollens being measured from ambient atmosphere during any of the mentioned measurement periods. Instead of these images it make more sense to show images of ruptured pollens either collected from ambient atmosphere or from laboratory studies, which could further support their argument of presence of pollen fragments < 2.5 $\mu$m in size."

Response to reviewer #2 specific remark 1: We carried-out microscopic imaging to determine diameters of pollens that we chemically profiled in this study. These images

were not taken to support any of the ambient PM measurements. To clearly convey the purpose of taking microscopic images we changed our wording as below.

"In the introduction section page 3, lines 31- 36 reads as "Our central objectives were...... iii) determine intact pollen diameters and chemically profile regionally-important pollen types (red oak, pin oak, cotton ragweed, giant ragweed and corn) for use in source apportionment, and iv) estimate pollen and fungal spore contributions to PM mass by way of chemical mass balance (CMB) modelling.

Moreover we incorporated the purpose of doing microscopy measurements in the method, section 2.5. Now page 6, line 16 reads as "Pollen images were taken to determine pollen grain diameters using a Zeiss LSM 710 fluorescence microscope (Carl Zeiss Microscopy GmbH, 07745 Jena, Germany) following PÓğhlker et al. (2012), and IX-81 inverted microscope (Olympus Corporation, Tokyo, Japan).

We agree with the reviewer that these microscopy measurements were not used in this study other than to visualize the pollen size and shape thus we moved the images of pollens (Figure 1) to the supplementary information Figure S1. The laboratory studies as suggested by the reviewer that demonstrated pollens releasing fragments of < 2.5 $\mu$m described in the introduction, page 2, lines 22-24 as "In rainy conditions, pollen grains absorb water, osmotically rupture, and release cytoplasmic starch granules (D'Amato et al., 2007). Microscopy studies have shown that intact birch pollens of 22 $\mu$m in size can rupture and release around 400 starch granules (Staff et al., 1999) ranging from 0.03 - 4 $\mu$m (D'Amato et al., 2007)."

Reviewer #2 specific remark 2: "In Fig. 2, PM2.5-10 mass on April 17 and 18 appears to be zero. But there is glucose detected in this size fraction (Fig. 2c). How is this possible?" ?

Response to reviewer #2 specific remark 2: We agree with the reviewer that coarse PM levels on April 17 and 18 are not visible in the Figure 1b (previously, Figure 2b). The coarse PM concentrations during those days were very low, compared to other days.

In particular, as indicated in SI Table S4, during April 17 and 18 PM10-2.5 was below our detection limit (<0.03 $\mu$g m-3). When these filters were subjected to more sensitive chemical analysis we obtained chemical measurements in units of ng m-3. The apparent discrepancy on 17 April and 18 is due to the carbohydrate measurement method being much more sensitive and having lower detection limits than mass measurements done by weighing.

Reviewer #2 specific remark 3: "Page 5, L 27: Correct as Biomarkers."

Response to reviewer #2 specific remark 3: We agree with the reviewer and in response we revised the text in page 5, line 27. The text in page 5, line 32 now reads as " Biomarkers were analyzed. . .. . .."

Reviewer #2 specific remark 4: "Page 7, L 25: "Rain also affected the distribution of particles between the fine and coarse modes, with 48±11 % of PM10 was less than 2.5 $\mu$m on rainy days compared to 80±13 % on dry days". This sentence is confusing. Is the author mentioning about %contribution of PM2.5 in PM10 during wet and dry days?"

Response to reviewer #2 specific remark 4: We agree with the reviewer and to avoid the confusion we revised the text on page 7, line 25. Now the revised text in page 8, line 5 reads as "Rain also affected the distribution of particles between the fine and coarse modes. PM2.5 contributed 48±11 % of PM10 on rainy days compared to 80±13 % on dry days."

Reviewer #2 specific remark 5: "Page 8, L 23: "passive release of larger pollen particles ranging 2.5–10 $\mu$m during others". What could be these larger pollen particles released passively during dry days? The microscopic images shows only pollens > 20 $\mu$m."

Response to reviewer #2 specific remark 5: We thank the reviewer for pointing this and in response we revised the text in page 9 line 6 to read as "Together, these data suggest release of pollen fragments less than 2.5 $\mu$m during some rain events (2–4

[Figure]

May) and the passive release of some pollen particles in the coarse particle size range during others (9 May)."

Reviewer #2 specific remark 6: "Page 9, L 1: From the ratio of glucose and sucrose the authors have related the pollens present on May 9 to that of red oak. Is there any other evidence to support this? The authors have mentioned in section 3.1 that the carbohydrate distribution in pollens are likely to differ with change in environmental factors. Hence it is difficult to relate the pollens to any particular type only based on carbohydrate ratio. Also the authors have not reported any such match in carbohydrate ratios on any other days or in coarse mode PM fraction."

Response to reviewer #2 specific remark 6: We have related the ambient carbohydrate measurements of May 2 to oak pollen profiles. May 2 has exceptionally high pollen tracer concentrations and these sugar ratios matched well with red oak profile. Also, oak trees are abundant in the Eastern Iowa and they are known to release pollen during springtime. For clarity we have revised the Page 9, lines 19-21 to read as below. With the information on ambient carbohydrate measurements and pollen profiles, which were done parallelly in spring 2013, our best approximation is that pollens are coming from red oak. We agree that further microscopy measurements would be useful to confirm this, but are beyond the scope of this study, as discussed in response to reviewer 2 general comment. "On 2 May, the relative ratios of glucose and sucrose (normalized to fructose) in fine PM were 1.4 and 2.5, respectively, close to the ratios of red oak (1.2 and 2.1, respectively). Oak trees are abundant in Eastern Iowa and a prominent pollen type in the springtime, making oak a likely (but unconfirmed) source of pollens in fine PM."

Reviewer #2 specific remark 7: "Page 9, L 16: "shift in glucose size distribution to 34% in the fine mode". It is not actually the size distribution, instead the relative contribution of glucose in fine fraction increases."

Response to reviewer #2 specific remark 7: We thank the reviewer for pointing this out.

Page 10, lines 2-4 reads as follows "The single late-summer rain event on 22 August coincided with an increase in fine mode glucose concentration and an increase of the fine PM fraction of glucose to 34%, compared to 16% on dry days."

Reviewer #2 specific remark 8: "Page 9, L 23-24: "Daily concentrations of coarse mode concentrations of two fungal spore tracers—fungal sugar mannitol and the fungal cell wall component glucan—were significantly correlated with daily average temperature (rs>0.4, p<0.05)." This statement appears to be significant only for mannitol and not glucan. In L 27, The authors have reported an increase in fungal spore tracer level with increase in temperature. But no significant increase in glucan level can be seen in the graph (Fig. 4b)."

Response to reviewer #2 specific remark 8: To be clear about the statistical results we revised our sentence in page X, lines 10-12 to read: "Daily coarse mode fungal spore tracer concentrations significantly correlated with daily average temperature: fungal sugar mannitol and temperature (rs=0.7, p<0.001) and the fungal cell wall component glucan and temperature— (rs=0.4, p=0.04)." The statistical correlations performed here are Spearman's rank correlations, a non-parametric correlation test that use ranks of the measurements as stated in section 2.7. Due to the differences in resulted rs (rs = 0.7 vs rs = 0.4) values and the significance of the correlations of temperature with mannitol and glucans it is understandable that reviewer 2 having hard time visualizing trends of temperature and glucans with the graphs where we have plotted both coarse and fine mode concentrations together. Thus we revised our wording as above to clearly mention the statistical results.

Reviewer #2 specific remark 9: "Page10 L 1: "Fungal spore tracer levels dropped on days when rain fell (e.g. 23 April, 2 May), due to particle removal by wet deposition". The drop in tracer levels is visible only in coarse PM. The mannitol concentration in fine fraction actually shows an increase on 23 April as compared to the previous day without rain. Same is for glucan on 23 April. Also the relative contribution of fungal tracer in fine PM is high on these rainy days as compared to other dry days. Hence

this statement is true only for tracer levels in coarse PM."

Response to reviewer #2 specific remark 9: We agree with the reviewer and accordingly we changed the text in page 10, line 1. Now page 10 line 25 reads as "Fungal spore tracer levels in coarse PM dropped on days when rain fell (e.g. 23 April, 2 May), due to particle removal by wet deposition."

Reviewer #2 specific remark 10: "Page 10 L19: "which suggests an alternative non-fungal source of glucan". Pollen is a likely source'. If glucan can have other non-fungal source including pollen, then how can it be used as tracer for fungal spore. The high increase in glucan in PM10 on 22 April (Page 9, L 26) might be due to pollens which are generally released due during higher temperature"

Response to reviewer #2 specific remark 10: Assessment of ambient fungal glucan level is very important as they are directly associated with negative health impacts. In response to this reviewer comment we expanded our discussion in page 11, lines 6-21 to read: "Coarse mode glucan concentrations in late summer were neither correlated with temperature (rs=0.01, p=1) nor mannitol (rs=0.2, p=0.3). Mannitol concentrations and fungal spore counts have spatial and seasonal differences from one another (Bauer et al., 2008), likely due to differences in mannitol emission per spore across fungal types (Elbert et al., 2007; Bauer et al., 2008) and/or mannitol concentrations in spores from within a species (e.g. ascomycetes releases ascospores during sexual reproduction and conidia during asexual reproduction (Nauta and Hoekstra, 1992)). The glucan content in fungal cell walls also vary with the fungal species (Foto et al., 2004). Collectively, these differences could give rise to weak or negligible correlations of ambient mannitol and glucan concentrations. Alternatively, non-fungal sources of either mannitol or glucans would confound their correlation. For instance, higher plants and some algae contain mannitol in their structure (Loescher et al., 1992; Shen et al., 1997). Ragweed pollens contain glucans (Foto et al., 2004), is a possible glucan source in late summer when ragweed pollens are prevalent and glucans significantly correlate with sucrose (rs=0.5, p=0.04). Alternatively glucans may have derived from

bacterial cells (McIntosh et al., 2005; Rylander and Lin, 2000), although their correlation was not significant (rs=0.4, p=0.1). Although glucans appear to have been influenced by bacterial and pollen levels in addition to fungi, the assessment of their ambient concentrations remains important, because they are immunostimulants that negatively impact human health (Thorn, 2001; Bonlokke et al., 2006)."

Reviewer #2 specific remark 11: "Section 3.5: It is interesting to note that 92% of bacterial endotoxin is present in coarse fraction of PM. Generally one would expect bacterial endotoxin to be abundant in fine fraction of PM as bacteria are smaller in size. Authors have not given any satisfactory explanation for this low concentration of endotoxin in fine PM."

Response to reviewer #2 specific remark 11: We thank the reviewer for pointing this out. In response, we added the requested explanation to our discussion of the size distribution of bacterial endotoxins. Now page 11, line 32 reads as "On average, 92±5 % of PM10 endotoxins were in the coarse mode (Figure 3c). The distribution of bacterial endotoxins as well as bacterial cells towards larger particles has been demonstrated previously (Nilsson et al., 2011; Monn et al., 1995; Shaffer and Lighthart, 1997). Such observations reflect the association of bacteria with particles prominent in coarse mode such as plant parts, animal parts, soil, spores or pollen surfaces (Jones and Harrison, 2004; Shaffer and Lighthart, 1997). In addition, it has been suggested that bacteria settled on particles are more likely to survive in the atmosphere compared to a single bacterium (Lighthart et al., 1993)."

Reviewer #1 specific remark 12: "Page 13, L 21-22: "The release of pollens, fungal spores, and Gram negative bacteria in fine particles during rain events, as observed surrounding spring and late-summer rain events in Iowa, has the potential to influence human health". This statement cannot be generalized at least for gram-negative bacteria during spring season where no increase in bacterial endotoxin in fine PM was observed during rain event."

Response to reviewer #2 specific remark 12: We agree with the reviewer on this comment. In response we revised the text. Page 14, line 22 now reads as "In general, the release of pollens, fungal spores, and Gram-negative bacteria in fine particles during rain events in Iowa, have the potential to influence human health"

Reviewer #2 specific remark 13: "Page 14, L 21-22: "Airborne fungal spore tracers, however, were suppressed by spring rain and increased in concentration following rain events". I feel this line contradicts the statement given in page 13, L 21, 'The release of pollens, fungal spores, and Gram negative bacteria in fine particles during rain events, as observed surrounding spring and late-summer rain events in Iowa, has the potential to influence human health".

Response to reviewer #2 specific remark 13: We agree with this reviewer comment and page 14, line 21-23 is revised. Page 16, line 6 now read as "Airborne fungal spore tracers in coarse PM fraction, however, were suppressed by spring rain and increased in concentration following rain events"

References

Bauer, H., Claeys, M., Vermeylen, R., Schueller, E., Weinke, G., Berger, A., and Puxbaum, H.: Arabitol and mannitol as tracers for the quantification of airborne fungal spores, Atmos. Environ., 42, 588-593, doi: 10.1016/j.atmosenv.2007.10.013, 2008.

Bonlokke, J. H., Stridh, G., Sigsgaard, T., Kjærgaard, S. K., Löfstedt, H., Andersson, K., Bonefeld-Jørgensen, E. C., Jayatissa, M. N., Bodin, L., and Juto, J.-E.: Upper-airway inflammation in relation to dust spiked with aldehydes or glucan, Scand. J. Work. Environ. Health, 374-382, doi, 2006.

D'Amato, G., Liccardi, G., and Frenguelli, G.: Thunderstorm'ÄĎasthma and pollen allergy, Allergy, 62, 11-16, doi: 10.1111/j.1398-9995.2006.01271.x, 2007.

Elbert, W., Taylor, P., Andreae, M., and Pöschl, U.: Contribution of fungi to primary biogenic aerosols in the atmosphere: wet and dry discharged spores, carbohydrates,

and inorganic ions, Atmos. Chem. Phys., 7, 4569-4588, doi: 10.5194/acp-7-4569-2007, 2007.

Foto, M., Plett, J., Berghout, J., and Miller, J. D.: Modification of the Limulus amebocyte lysate assay for the analysis of glucan in indoor environments, Anal. Bioanal. Chem., 379, 156-162, doi: 10.1007/s00216-004-2583-4, 2004.

Jones, A. M., and Harrison, R. M.: The effects of meteorological factors on atmospheric bioaerosol concentrations—a review, Sci. Total Environ., 326, 151-180, doi: 10.1016/j.scitotenv.2003.11.021, 2004.

Lighthart, B., Shaffer, B. T., Marthi, B., and Ganio, L. M.: Artificial wind-gust liberation of microbial bioaerosols previously deposited on plants, Aerobiologia, 9, 189-196, doi: 10.1007/BF02066261, 1993.

Loescher, W. H., Tyson, R. H., Everard, J. D., Redgwell, R. J., and Bieleski, R. L.: Mannitol Synthesis in Higher Plants Evidence for the Role and Characterization of a NADPH-Dependent Mannose 6-Phosphate Reductase, Plant Physiol., 98, 1396-1402, doi: 10.1104/pp.98.4.1396 1992.

McIntosh, M., Stone, B., and Stanisich, V.: Curdlan and other bacterial $(1\rightarrow 3)$-$\beta$-D-glucans, Appl. Microbiol. Biotechnol., 68, 163-173, doi: 10.1007/s00253-005-1959-5, 2005.

Monn, C., Braendli, O., Schaeppi, G., Schindler, C., Ackermann-Liebrich, U., Leuenberger, P., and Team, S.: Particulate matter< 10 $\mu$m (PM10) and total suspended particulates (TSP) in urban, rural and alpine air in Switzerland, Atmos. Environ., 29, 2565-2573, doi: 10.1016/1352-2310(95)94999-U, 1995.

Nauta, M., and Hoekstra, R.: Evolution of reproductive systems in filamentous ascomycetes. I. Evolution of mating types, Heredity (Edinb.), 68, 405-410, doi: 10.1038/hdy.1992.60, 1992.

Nilsson, S., Merritt, A., and Bellander, T.: Endotoxins in urban air in Stockholm, Swe-

den, Atmos. Environ., 45, 266-270, doi: 10.1016/j.atmosenv.2010.09.037, 2011.

Pöhlker, C., Huffman, J., and Pöschl, U.: Autofluorescence of atmospheric bioaerosols–fluorescent biomolecules and potential interferences, Atmos. Meas. Tech., 5, 37-71, doi: 10.5194/amt-5-37-2012, 2012.

Rylander, R., and Lin, R.-H.: $(1 \rightarrow 3)$-$\beta$-d-glucan—relationship to indoor air-related symptoms, allergy and asthma, Toxicology, 152, 47-52, doi, 2000.

Shaffer, B. T., and Lighthart, B.: Survey of culturable airborne bacteria at four diverse locations in Oregon: urban, rural, forest, and coastal, Microb. Ecol., 34, 167-177, doi: 10.1007/s002489900046, 1997.

Shen, B., Jensen, R. G., and Bohnert, H. J.: Increased resistance to oxidative stress in transgenic plants by targeting mannitol biosynthesis to chloroplasts, Plant Physiol., 113, 1177-1183, doi: 10.1104/pp.113.4.1177 1997.

Staff, I., Schäppi, G., and Taylor, P.: Localisation of allergens in ryegrass pollen and in airborne micronic particles, Protoplasma, 208, 47-57, doi: 10.1007/BF01279074, 1999.

Thorn, J.: Seasonal variations in exposure to microbial cell wall components among household waste collectors, Ann. Occup. Hyg., 45, 153-156, doi: 10.1093/a

---

## Author Response (AR2)

Response to Co-Editor Comments
Received: 04 January 2017
Atmospheric Chemistry and Physics – acp-2016-622

Influence of Rain on the Abundance of Bioaerosols in Fine and Coarse Particles

Chathurika M. Rathnayake[1], Nervana Metwali[2], Thilina Jayarathne[1], Josh Kettler[1], Yuefan Huang[1], Peter S. Thorne[2,3], Patrick T. O'Shaughnessy[2,3],  and Elizabeth A. Stone[1]

[1] Department of Chemistry, University of Iowa, Iowa City, 52242, United States.
[2] Occupational and Environmental Health, University of Iowa City, 52242, United States.
[3] Civil and Environmental Engineering, University of Iowa City, 52242, United States.

*Corresponding author phone: +1-319-384-1863, fax: +1-319-335-1270; email: betsy-stone@uiowa.edu

**Co-editor (Asst. Prof. Sachin S. Gunthe) comments**

Co-editor general comment: *"I have gone through the manuscript now. And I am quite happy with the revised version of the manuscript. The comments/suggestions given by the Reviewers' have been properly addressed/implemented in the manuscript. I, however, still have the concern about addressing the size distribution in the manuscript text.*

I still have few concerns about giving the emphasis to the importance of the size distribution, which is not measured by the authors in the manuscript. Instead only PM 2.5 and PM10 have been sampled. For example in abstract authors have mentioned that they have assessed size by daily measurements. The last sentence of the abstract again mentions that study defines changes to the size distribution and concentration. It has been mentioned on numerous occasions.

Also revise the introduction: Authors are emphasizing the importance of measurements of size distribution, which they have not done.

Further, in my opinion the details of each type of bioaerosol for given season could be discussed rather than separating by bioaerosol type. For example Late summer: discuss the details of all bioaerosol type for this season and so on. However, this could be a personal choice and leave it to the Authors' to decide.

Please revise the manuscript thoroughly by clearly mentioning that it is fine and coarse mode measurements. Page 8 line 6; Page 8 line 33; page 9 line 4 and many more.*"*

**Response to co-editor general comment:** We thank the co-editor for his input. We have revised the manuscript in response to each comment point-by-point below.

In order to address the co-editor comment on the abstract we revised the last sentence of the abstract in page1, lines 26-27 to read "Overall, this study defines changes to the fine and coarse mode distribution of PM pollens, fungal spores, ..."

To address the comment on the introduction we revised page 3, lines 28-30 to read as follows: " Measurement of these bioaerosol tracers allows for the evaluation of the atmospheric concentrations and fine and coarse mode distributions of pollens, fungal spores, and Gram-negative bacteria."

We also re-worded our objectives in page 3, line 31 to page 4 line 4. The text in page 3, line 32 to page 4 line 5 now reads as: "Our central objectives were *i*) to assess temporal variations in pollens, fungal spores and endotoxin concentrations and their distribution across fine ($PM_{2.5}$) and coarse ($PM_{10-2.5}$) size modes, *ii*) evaluate environmental conditions including rain and temperature that lead to high bioaerosol levels and decreases in their size from $PM_{10-2.5}$ to $PM_{2.5}$, *iii*) determine intact pollen diameters and chemically profile regionally-important pollen types (red oak, pin oak, cotton ragweed, giant ragweed and corn) for use in source apportionment, and *iv*) estimate pollen and fungal spore contributions to PM mass by way of chemical mass balance (CMB) modelling. The outcomes of this study include an improved understanding of changes in ambient bioaerosol concentrations and distributions across fine and coarse size modes in response to rain events and their contributions to PM mass."

We appreciate the co-editor suggestion on the organization of the discussion of bioaerosols, but prefer to maintain our discussion organized by bioaerosol type.

To address co-editor last comment on the size distribution of bioaerosols we revised the manuscript as summarized below:

[revised manuscript text omitted]

---

## Author Response (AR3)

Response to Editor Comments

Atmospheric Chemistry and Physics – acp-2016-622

Influence of Rain on the Abundance of Bioaerosols in Fine and Coarse Particles
Chathurika M. Rathnayake[1], Nervana Metwali[2], Thilina Jayarathne[1], Josh Kettler[1], Yuefan Huang[1], Peter S. Thorne[2,3], Patrick T. O'Shaughnessy[2,3],  and Elizabeth A. Stone[1]

[1] Department of Chemistry, University of Iowa, Iowa City, 52242, United States.
[2] Occupational and Environmental Health, University of Iowa City, 52242, United States.
[3] Civil and Environmental Engineering, University of Iowa City, 52242, United States.

*Corresponding author phone: +1-319-384-1863, fax: +1-319-335-1270; email: betsy-stone@uiowa.edu

**Editor comments:**

Dear Authors,

I am now happy to read the manuscript. Further, I am not really sure of the terminologies and definitions you have used about "fine particulate matter (PM2.5), coarse PM (PM10-2.5) and PM10 (as the combination of PM2.5 and PM10-2.5)". Either you try to provide the reference for this somewhere in the methodology section or please consider this making as PM2.5 (particulate matter <= 2.5 micro meter) and PM10 (particulate matter <= 20.0 micrometer) for simplicity.

Overall I am now satisfied with the manuscript and I congratulate all the authors of the manuscript.

With best wishes,

Sincerely,

Sachin

**Response to Editor comment:** We thank the editor for their suggestion to improve the clarity of the manuscript.

The quoted text appears in the abstract of the paper, which we have revised to read: "
[revised manuscript text omitted]